# Comparative genetic screens in human cells reveal new regulatory mechanisms in WNT signaling

Andres M Lebensohn[1,2], Ramin Dubey[1,2], Leif R Neitzel[3], Ofelia Tacchelly-Benites[4], Eungi Yang[4], Caleb D Marceau[5], Eric M Davis[6], Bhaven B Patel[1,2], Zahra Bahrami-Nejad[7], Kyle J Travaglini[1], Yashi Ahmed[4], Ethan Lee[3], Jan E Carette[5]*, Rajat Rohatgi[1,2]*

[1]Department of Biochemistry, Stanford University School of Medicine, Stanford, United States; [2]Department of Medicine, Stanford University School of Medicine, Stanford, United States; [3]Department of Cell and Developmental Biology, Vanderbilt University Medical Center, Nashville, United States; [4]Department of Molecular and Systems Biology and the Norris Cotton Cancer Center, Geisel School of Medicine at Dartmouth College, Hanover, United States; [5]Department of Microbiology and Immunology, Stanford University School of Medicine, Stanford, United States; [6]Department of Molecular Cellular, and Developmental Biology, University of Colorado, Boulder, Boulder, United States; [7]Department of Chemical and Systems Biology, Stanford University School of Medicine, Stanford, United States

*For correspondence: carette@stanford.edu (JEC); rrohatgi@stanford.edu (RR)

**Competing interests:** The authors declare that no competing interests exist.

**Abstract** The comprehensive understanding of cellular signaling pathways remains a challenge due to multiple layers of regulation that may become evident only when the pathway is probed at different levels or critical nodes are eliminated. To discover regulatory mechanisms in canonical WNT signaling, we conducted a systematic forward genetic analysis through reporter-based screens in haploid human cells. Comparison of screens for negative, attenuating and positive regulators of WNT signaling, mediators of R-spondin-dependent signaling and suppressors of constitutive signaling induced by loss of the tumor suppressor adenomatous polyposis coli or casein kinase 1$\alpha$ uncovered new regulatory features at most levels of the pathway. These include a requirement for the transcription factor AP-4, a role for the DAX domain of AXIN2 in controlling $\beta$-catenin transcriptional activity, a contribution of glycophosphatidylinositol anchor biosynthesis and glypicans to R-spondin-potentiated WNT signaling, and two different mechanisms that regulate signaling when distinct components of the $\beta$-catenin destruction complex are lost. The conceptual and methodological framework we describe should enable the comprehensive understanding of other signaling systems.

## Introduction

Cellular signaling systems have evolved complex circuitry involving multiple layers of regulation, making their comprehensive characterization a major challenge. Forward genetics in model organisms has been a general and unbiased way to identify new components in signaling pathways and to map their connectivity. However, since signaling pathways have often diverged between humans and these simpler model systems, their analysis in human cells becomes an important goal. Indeed, our ability to identify the best therapeutic strategy or to predict the effectiveness of drugs targeting

**eLife digest** When an embryo is developing, its cells must communicate with one another to coordinate the processes that shape the body's tissues and organs. Cells often communicate by releasing signaling molecules that engage with proteins called receptors on the surface of other cells. This triggers a series of events that sends the signal along a "pathway" of biochemical reactions inside the receiving cell and leads to the activation of genes. One such signaling pathway is triggered by the WNT proteins and is used extensively in all animals. The WNT pathway instructs cells to grow and divide, establishes the identity of specific cell types and maintains populations of stem cells that can regenerate tissues in adulthood as well. The WNT pathway must be carefully regulated because various types of cancer can develop if the pathway becomes too active.

Some signaling pathways are well conserved between different animals. Many genetic studies into the WNT pathway have focused on animals that are easier to work with in the laboratory, like worms or flies. However, there may be differences in the way these pathways are regulated between these model animals and humans. Therefore, to understand how the WNT pathway operates in humans, it was important to study it in human cells too.

Lebensohn et al. have now carried out a series of genetic screens in human cells that contain only one copy of each gene instead of the usual two. These cells – referred to as haploid cells – are ideal for genetic studies because only a single copy of a gene has to be disrupted in order to analyze the consequences of that gene's loss. The screens searched for genes that regulate WNT signaling: those that keep the pathway "off" in the absence of WNT and those that turn the pathway "on" in response to WNT.

By comparing the outcomes of these screens, Lebensohn et al. identified previously unknown regulators and uncovered new roles for known regulators of the WNT pathway. For instance, a regulator called TFAP4, which had not previously been linked to the pathway, was shown to activate WNT signaling. In another case, enzymes that make molecules called glycophosphatidylinositol anchors, and cell-surface proteins that are modified with those anchors, were found to amplify WNT signaling.

Lebensohn et al. also identified genes that were needed to sustain the uncontrolled WNT signaling in cells that carried cancer-causing mutations in this pathway. Further studies could now explore if drugs can target these genes, or the molecules encoded by them, to treat cancers in which the WNT pathway is excessively activated. Other studies could also use the same methods to explore more signaling pathways and gain new insights into important biological processes in human cells.

specific proteins is often hampered by an incomplete understanding of signaling circuitry in human cells (*Lito et al., 2013*). Recent methodological advances have enabled the interrogation of biological processes in human cells through powerful genome-wide screens that overcome many of the limitations associated with previous platforms (*Carette et al., 2009*; *Gilbert et al., 2014*; *Shalem et al., 2014*; *Wang et al., 2014*). Yet, inferring functional relationships in complex pathways from such screens remains a major obstacle that has only recently began to be addressed (*Bassik et al., 2013*; *Blomen et al., 2015*; *Parnas et al., 2015*; *Wang et al., 2015*).

Genetics has long relied on the use of sensitized backgrounds, modifier screens and synthetic effects to uncover the myriad layers of regulation in signaling pathways. We reasoned that one way to discover both epistatic relationships on a genome scale and unique context-specific requirements would be through the quantitative comparison of genome-wide screens in which the pathway is activated by different ligands, and of suppressor screens following targeted disruption of critical nodes. We took advantage of two methodologies to conduct a systematic genetic analysis of WNT signaling in human cells: forward genetics in haploid cells using gene trap (GT)-based insertional mutagenesis (*Carette et al., 2009*), and targeted genome engineering by clustered regularly-interspaced short palindromic repeats (CRISPR)/CRISPR-associated protein 9 (Cas9) (*Cong et al., 2013*; *Mali et al., 2013*).

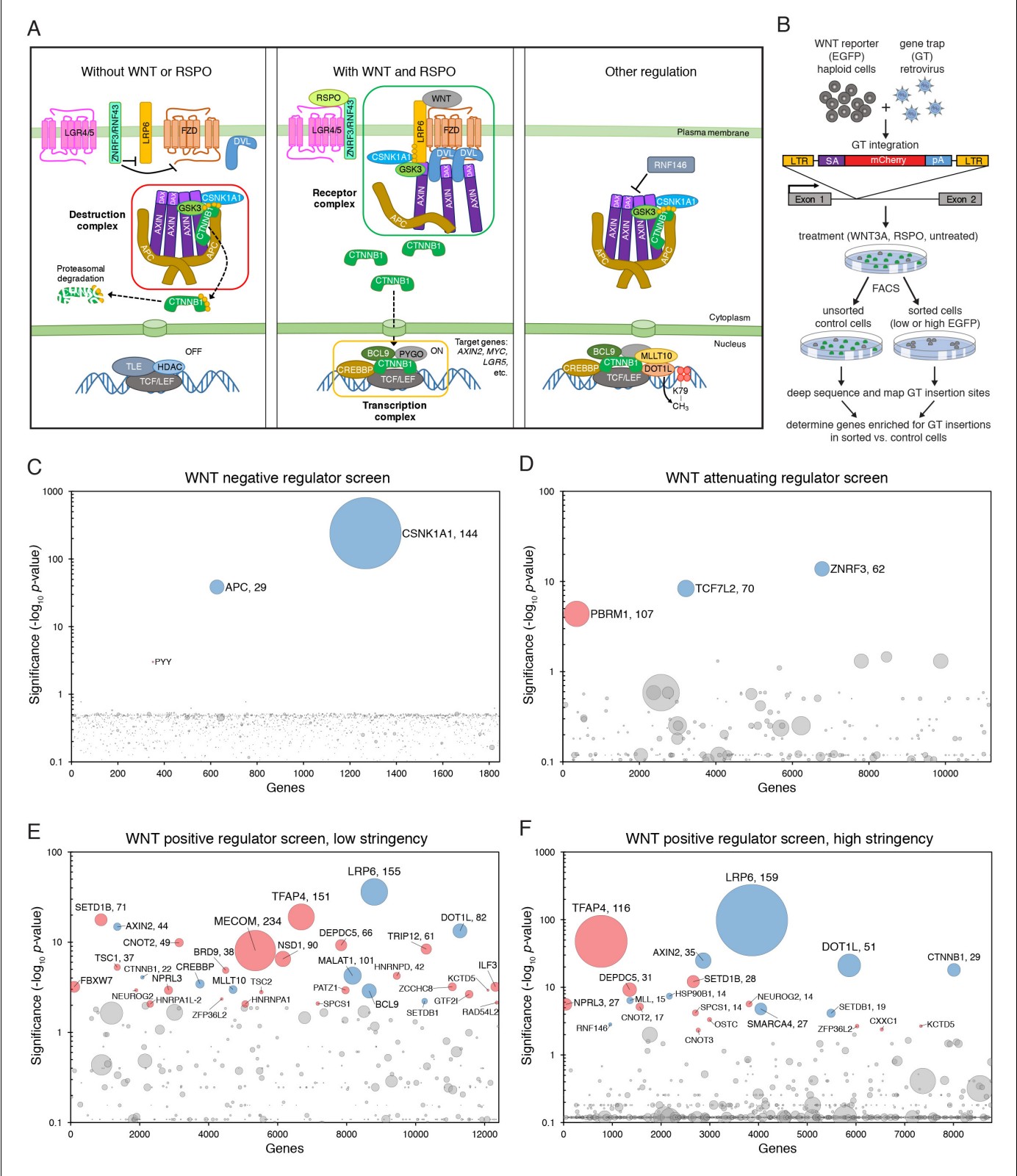

**Figure 1.** Reporter-based, forward genetic screens in haploid human cells identify negative, attenuating and positive regulators of WNT signaling. (**A**) Schematic model of canonical WNT signaling, highlighting the main pathway components and regulatory events in the absence (left panel) and presence (center panel) of ligands, and other known regulators relevant to this work (right panel). When the pathway is off, the transcriptional co-activator β-catenin (CTNNB1) is constitutively targeted for proteasomal degradation by the destruction complex, composed of the scaffold proteins

*Figure 1 continued on next page*

Figure 1 continued

adenomatous polyposis coli (APC) and AXIN, and the kinases glycogen synthase kinase 3 (GSK3) and casein kinase 1α (CSNK1A1). The T-cell-specific transcription factor (TCF)/lymphoid enhancer-binding factor (LEF) family of transcription factors, together with transducin like enhancer of split (TLE) and histone deacetylases (HDAC), repress WNT target genes. Binding of WNT to its co-receptors frizzled (FZD) and low-density lipoprotein receptor-related protein 6 (LRP6) leads to the assembly of a receptor complex that inactivates the destruction complex through a mechanism involving recruitment of AXIN by LRP6 and the adapter protein dishevelled (DVL). Consequently, CTNNB1 accumulates in the cytoplasm, translocates to the nucleus and promotes WNT target gene transcription in cooperation with TCF/LEF and other co-activators such as CREB-binding protein (CREBBP) and B-cell CLL/lymphoma 9 protein (BCL9). R-spondins (RSPOs) are secreted proteins that potentiate the response of stem cells to WNT ligands by blocking the degradation of FZD and LRP6 receptors. RSPO binds to leucine-rich repeat-containing G-protein-coupled receptors (LGRs) and neutralizes two transmembrane E3 ubiquitin ligases, ZNRF3 and RNF43, that clear WNT receptors from the cell surface. Other regulatory mechanisms include modulation of AXIN levels by the poly ADP-ribosylation-dependent E3 ubiquitin ligase RNF146, and recruitment of DOT1L and MLLT10, two proteins involved in histone H3 K79 methylation, to WNT target genes. (B) Schematic of WNT reporter-based forward genetic screens in haploid human cells using a GT-bearing retrovirus for mutagenesis, followed by phenotypic enrichment by FACS. LTR, long terminal repeats; SA, splice acceptor; pA, polyadenylation signal. (C–F) Circle plots depicting genes enriched for GT insertions in screens for negative (C), attenuating (D) and positive (E and F) regulators of WNT signaling. Two independent screens for positive regulators were performed at low (E) and high (F) selection stringencies by sorting for cells with the lowest 10% and 2% WNT reporter fluorescence, respectively. The y-axis indicates the significance of GT insertion enrichment in the sorted vs. the control cells (expressed in units of $-\log_{10}$FDR-corrected $p$-value) and the x-axis indicates genes (in random order) for which GT insertions were mapped in the sorted cells. Genes with FDR-corrected $p$-value<0.01 are labeled and colored in light blue if they encode a known pathway component, or in pink if their product has not been previously implicated as a regulator of canonical WNT signaling. The diameter of each circle is proportional to the number of unique inactivating GT insertions mapped in the sorted cells, which is also indicated next to the gene name for the most significant hits with FDR-corrected $p$-values$<10^{-4}$.

The following figure supplement is available for figure 1:

**Figure supplement 1.** Characterization of HAP1-7TGP, a haploid human cell line harboring a WNT-responsive fluorescent reporter (A–D), and depiction of FACS gates and phenotypic enrichment during various reporter-based forward genetic screens (E–H).

The WNT pathway is a fundamental signaling system that plays central roles in embryonic development, regeneration and cancer (reviewed in *Hoppler and Moon, 2014*). During development, WNT signaling orchestrates transcriptional programs that regulate cell proliferation and survival, cell fate determination, and tissue patterning. In adults, WNT signaling is instrumental in defining stem cell niches in multiple organs, which maintain tissue homeostasis during routine turnover or following injury. Overactive WNT signaling can be oncogenic, driving both the initiation and maintenance of various types of cancer, most notably colorectal cancer (CRC).

While the pathway has been studied intensively (we provide a snapshot in *Figure 1A* and refer readers to the legend for details), critical steps remain poorly understood even 34 years after the discovery of '*int1*', as mammalian WNT was initially called (*Nusse and Varmus, 1982*). The complex circuitry of the pathway may mask unknown regulatory mechanisms overlaid on the core module, making it an ideal system for an in-depth, methodical genetic dissection, extending a rich tradition of genetic studies (*Nüsslein-Volhard and Wieschaus, 1980*). Known pathway components would serve to benchmark any new discoveries, and new discoveries would likely have important therapeutic implications due to the pathway's direct relevance to stem cell biology and cancer.

We initially probed the intact pathway through genome-wide, forward genetic screens for negative, attenuating and positive regulators to define the basic set of genes required for signaling in haploid human cells. We identified many of the known core pathway components and found a new requirement for the transcription factor AP-4 (TFAP4). Unexpectedly, these screens uncovered a dominant allele of *AXIN2* that dissociated β-catenin (CTNNB1) protein stabilization, considered the decisive event in WNT signaling, from its transcriptional activity. To find factors required for amplification of WNT responses by R-spondins (RSPOs, *Figure 1A*), we devised a sensitized screen for RSPO-enhanced WNT signaling and uncovered a requirement for the glycophosphatidylinositol (GPI) anchor biosynthetic machinery and glypicans. Finally, we searched for mutations that could suppress constitutive signaling in cells with compromised function of the CTNNB1 destruction complex, recapitulating the most common defect in oncogenic WNT signaling. Suppressor screens in cells lacking adenomatous polyposis coli (APC) or casein kinase 1α (CSNK1A1), the two rate-limiting negative regulators of the pathway in haploid cells, revealed two distinct mechanisms that regulate CTNNB1 abundance and transcriptional activity, respectively. One mechanism was specific to cells

lacking CSNK1A1, but not APC, suggesting that different components of the destruction complex have different functions in WNT signaling beyond their common function controlling CTNNB1 protein abundance.

Through a quantitative comparative analysis across seven screens, we confirmed epistatic relationships for known regulators and predicted them for new ones. The combined results of these screens provide a comprehensive resource for understanding the regulation of canonical WNT signaling.

## Results

### Forward genetic screens in haploid human cells identify negative, attenuating, and positive regulators of WNT signaling

A central goal of this project was to search for quantitative and context-specific regulators of WNT signaling in an unbiased and comprehensive manner. We adopted two design principles that exploited the flexibility of forward genetics in cultured human cells. First, as a means of phenotypic enrichment, we chose a fluorescence-based, quantitative transcriptional reporter of WNT signaling. Since WNT reporter fluorescence is a continuous readout, in contrast to digital readouts such as cell viability or the presence or absence of a phenotype, it enabled us to enrich for cells with enhanced or reduced signaling phenotypes by fluorescence activated cell sorting (FACS) with complete flexibility on the stringency of selection. Second, all screens were performed in a pooled format following genome-scale insertional mutagenesis using a GT-bearing retrovirus, which contains a strong splice acceptor site and can therefore disrupt genes when it integrates in either exons or introns. This mutagenesis method is untargeted, distinguishing it from approaches in which short hairpin RNAs or single guide RNAs (sgRNAs) are designed to perturb a pre-defined set of cistrons.

We constructed and thoroughly characterized a clonal haploid human cell line, hereafter called HAP1-7TGP, in which expression of enhanced green fluorescent protein (EGFP) is driven by an established WNT-responsive element containing the seven TCF/LEF-binding sites, minimal promoter and 5'UTR of the SuperTOPflash reporter (*Fuerer and Nusse, 2010*; *Figure 1—figure supplement 1A–1D*). While this construct has been used extensively to report on WNT responses, it may not mimic all endogenous regulatory sequences driving WNT target gene expression. In particular, the effects of proteins involved in modifying chromatin structure could differ between the reporter and endogenous target genes. Given these limitations, whenever possible we confirmed new regulatory mechanisms by measuring endogenous WNT target gene activity or assessing WNT-dependent phenotypes in model organisms.

To obtain saturating mutational coverage of the genome, we started our screens with 120 million HAP1-7TGP cells (or engineered derivatives thereof) mutagenized with the GT retrovirus, ensuring that the mutant cell population as a whole contained multiple (up to a few hundred) independent lesions in every gene (*Figure 1B*). The screens should therefore capture most genes involved in the phenotype being enriched for, except for genes required for the viability of haploid cells and genes with redundant function, since the probability of independent GT integrations disrupting redundant genes in the same haploid cell is vanishingly small. This limitation is inherent to all forward genetic screens that use random or untargeted mutagenesis.

After sequential rounds of FACS-based phenotypic enrichment and growth (HAP1-7TGP cells do not require WNT signaling for growth, enabling the propagation of cells with decreased or increased WNT signaling activity following phenotypic enrichment), we mapped retroviral integration sites at nucleotide resolution by deep sequencing an amplified library containing junctions between GTs and flanking genomic DNA (*Figure 1B*; see Materials and methods). Sequence reads from the sorted cells were compared to those from control, unsorted cells to identify genes enriched for GT insertions in the sorted cell population. Disruption of these genes would be expected to cause the phenotype used as the basis for selection.

We devised a genome-wide screen to identify the rate-limiting negative regulators of WNT signaling (i.e. genes whose disruption leads to constitutive pathway activity) in haploid human cells. We used FACS to sort mutagenized HAP1-7TGP cells with high WNT reporter activity in the absence of WNT ligand ('EGFP$^+$' gate in *Figure 1—figure supplement 1E*). Following two rounds of sorting (see Materials and methods), GT insertions in only three genes showed statistically significant (false

**Table 1.** Relative gene expression level of selected WNT pathway regulators in HAP1 cells. RPKM values from duplicate RNAseq datasets generated as described in Materials and methods from two different passages of WT HAP1 cells are shown. Groups of paralogues and genes with similar functions are shaded in alternating colors to facilitate comparisons.

| Gene | RPKM | | |
|------|-------------|-------------|---------|
| | Replicate 1 | Replicate 2 | Average |
| LGR4 | 160.61 | 174.69 | 167.65 |
| LGR5 | 0.02 | 0.00 | 0.01 |
| LGR6 | 0.02 | 0.00 | 0.01 |
| ZNRF3 | 30.90 | 33.30 | 32.10 |
| RNF43 | 0.12 | 0.08 | 0.10 |
| LRP5 | 55.92 | 64.38 | 60.15 |
| LRP6 | 109.51 | 121.08 | 115.30 |
| FZD1 | 19.57 | 18.85 | 19.21 |
| FZD2 | 19.56 | 21.09 | 20.33 |
| FZD3 | 48.02 | 55.82 | 51.92 |
| FZD4 | 19.60 | 22.10 | 20.85 |
| FZD5 | 31.85 | 34.52 | 33.19 |
| FZD6 | 33.53 | 31.95 | 32.74 |
| FZD7 | 13.89 | 14.89 | 14.39 |
| FZD8 | 4.02 | 4.48 | 4.25 |
| FZD9 | 3.66 | 2.80 | 3.23 |
| FZD10 | 10.40 | 9.86 | 10.13 |
| DVL1 | 73.91 | 69.61 | 71.76 |
| DVL2 | 51.74 | 48.80 | 50.27 |
| DVL3 | 88.84 | 90.25 | 89.54 |
| APC | 80.47 | 85.22 | 82.84 |
| APC2 | 2.94 | 3.69 | 3.32 |
| AXIN1 | 55.97 | 54.07 | 55.02 |
| AXIN2 | 10.04 | 12.51 | 11.27 |
| CSNK1A1 | 111.18 | 109.57 | 110.37 |
| GSK3A | 75.97 | 69.21 | 72.59 |
| GSK3B | 62.79 | 69.98 | 66.39 |
| TCF7L2 | 23.89 | 27.69 | 25.79 |
| LEF1 | 12.34 | 14.80 | 13.57 |
| CTNNB1 | 324.05 | 308.53 | 316.29 |
| CREBBP | 141.92 | 165.58 | 153.75 |
| PIGL | 4.07 | 4.51 | 4.29 |
| GPC4 | 209.39 | 229.86 | 219.63 |
| GPC6 | 13.88 | 14.90 | 14.39 |
| TFAP4 | 36.99 | 41.94 | 39.46 |
| SERBP1 | 721.65 | 698.99 | 710.32 |
| HUWE1 | 631.58 | 777.06 | 704.32 |

discovery rate (FDR)-corrected *p*-value<0.01) enrichment in the sorted cells: *CSNK1A1*, *APC* and *PYY* (*Figure 1C* and *Supplementary file 1*). We mapped 144 independent GT insertions in *CSNK1A1* and 29 in *APC*, showing that our mutagenesis had indeed targeted each gene multiple times. CSNK1A1 and APC are core components of the destruction complex that suppresses WNT signaling by promoting CTNNB1 degradation (*Figure 1A*); their identification as top hits reassured us that our screening strategy could identify important regulators of canonical WNT signaling. Genes encoding other known negative regulators of the pathway, such as *GSK3A* and *GSK3B* or *AXIN1* and *AXIN2*, presumably did not score as hits in this screen due to redundancy, as suggested by their expression profile in HAP1 cells (*Table 1*). We demonstrate later that AXIN1 and AXIN2 are indeed functionally redundant in HAP1 cells (*Figure 3—figure supplement 1B*). The fact that *APC,* but not *APC2,* was identified as a hit indicates that these two genes are not redundant in HAP1 cells, a conclusion that is supported by the relatively low expression level of *APC2* (*Table 1*).

To identify attenuating regulators of WNT signaling (i.e. genes whose disruption amplifies cellular responses to WNT ligands), we stimulated mutagenized HAP1-7TGP cells with a sub-saturating dose of WNT3A (12.5% WNT3A conditioned medium (CM), *Figure 1—figure supplement 1B*) and sorted for cells with the highest 2% EGFP fluorescence ('highest 2%' gate in *Figure 1—figure supplement 1F*). Following two rounds of FACS sorting, three genes were significantly enriched for GT insertions (*Figure 1D* and *Supplementary file 1*). *ZNRF3* was the top hit. Eliminating ZNRF3 or RNF43, two transmembrane E3 ubiquitin ligases, has been shown to amplify WNT signaling by increasing FZD and LRP6 levels on the cell surface (*Hao et al., 2012*; *Koo et al., 2012*; *Figure 1A*). Only *ZNRF3* is expressed at significant levels in HAP1 cells (*Table 1*), explaining why it was a hit in this screen. The second most significant hit of this screen was *TCF7L2*, encoding a TCF/LEF family transcription factor that can also function as an attenuating regulator of WNT target genes (*Tang et al., 2008*). Loss of either ZNRF3 or TCF7L2 is predicted to potentiate signaling responses rather than making them WNT-independent, explaining why these genes did not score in the negative regulator screen (*Figure 1C* and *Supplementary file 1*). These findings highlight one of the advantages of using a reporter with a graded output: different regulatory layers

in the pathway can be revealed by subtle alterations in selection conditions.

In a screen for positive regulators of canonical WNT signaling (i.e. genes whose disruption reduces signaling output), we stimulated mutagenized HAP1-7TGP cells with a near-saturating dose of WNT3A (50% WNT3A CM, *Figure 1—figure supplement 1B*) and enriched for cells with the lowest 10% reporter fluorescence ('lowest 10%' gate in *Figure 1—figure supplement 1G*) during two sequential rounds of FACS sorting and amplification. Thirty-three genes were significantly enriched for GT insertions in the sorted cells (*Figure 1E* and *Supplementary file 1*). These included genes encoding several known positive regulators of the pathway, such as the WNT co-receptor LRP6, components of the WNT transcription complex including CTNNB1, CREBBP and BCL9, and components of a histone H3 K79 methyltransferase complex including DOT1L and MLLT10 (*Figure 1A*). As expected, regulators with redundant expression profiles in HAP1 cells, such as FZDs or DVLs (*Table 1*), were not recovered in this screen. Increasing the stringency of selection by sorting for cells with the lowest 2% reporter fluorescence ('lowest 2%' gate in *Figure 1—figure supplement 1H*) did not change the results of the screen significantly (*Figure 1F* and *Supplementary file 1*); despite differences in the significance of GT insertion enrichment compared to the less stringent screen, the order of the top hits was generally maintained. Considering the multiple experimental steps involved, the results are remarkably reproducible. Henceforth, we refer to each of these two screens for positive regulators as the 'low stringency' (*Figure 1E*) and the 'high stringency' (*Figure 1F*) screen, respectively, and to both of them jointly as the 'WNT screens'.

The second most significant hit in both WNT screens for positive regulators, following *LRP6*, was *TFAP4* (*Figure 1E and F*, and *Supplementary file 1*), a gene encoding a transcription factor not previously implicated in regulation of canonical WNT signaling. The fourth and third most significant hit in the low (*Figure 1E*) and high (*Figure 1F*) stringency WNT screens, respectively, was *AXIN2*, encoding the CTNNB1 destruction complex scaffold AXIN2. It was perplexing to find *AXIN2* in a screen for *positive* regulators, since components of the destruction complex are *negative* regulators of the pathway, as illustrated by the presence of *APC* and *CSNK1A1* in our initial screen (*Figure 1C*). Experimental validation and analysis of both *TFAP4* and *AXIN2* follows in the two sections below.

These results establish that reporter-based, genome-wide forward genetic screens in haploid human cells are an effective way to identify many non-redundant components of signaling pathways. The versatility afforded by the combination of a reporter with a continuous fluorescence readout and FACS as a means of enrichment enables identification of functionally distinct classes of genes including negative, attenuating and positive regulators.

## TFAP4 regulates WNT signaling downstream of the CTNNB1 destruction complex

The second most significant hit in both screens for positive regulators of WNT signaling (*Figure 1E and F*, and *Supplementary file 1*), was the gene encoding the transcription factor TFAP4, outranked only by the gene encoding the WNT co-receptor LRP6. TFAP4 is a helix-loop-helix leucine zipper transcription factor and a target of MYC (*Jung and Hermeking, 2009*) that has been implicated in epithelial-to-mesenchymal transformation and metastasis in CRC (*Jackstadt et al., 2013*; *Shi et al., 2014*). Despite multiple reports correlating *TFAP4* expression and malignancy in gastrointestinal tumors (*Cao et al., 2009*; *Liu et al., 2012*; *Xinghua et al., 2012*), TFAP4 has not been previously implicated as a regulator of canonical WNT signaling.

We used CRISPR/Cas9 to generate two HAP1-7TGP cell lines, designated TFAP4$^{CR-1}$ and TFAP4$^{CR-2}$, the first of which lacks TFAP4 and the second of which produces a truncated protein product that retains the leucine zipper motif (see Materials and methods, *Figure 2C* and *Supplementary file 2*). We note that these and all the other cell lines generated using CRISPR/Cas9 and used in this work were isolated without any phenotypic selection and were genotyped by sequencing the single allele of the disrupted gene (see Materials and methods and *Supplementary file 2*). TFAP4$^{CR-1}$ and TFAP4$^{CR-2}$ cells showed a substantial reduction in WNT3A-induced expression of endogenous *AXIN2*, a target gene commonly used as a metric for pathway activity (*Figure 2A*). The defect in target gene induction correlated with the severity of the two mutant alleles of *TFAP4*. WNT3A-induced reporter activation in TFAP4$^{CR-1}$ cells could be rescued by re-expression of TFAP4 (*Figure 2B*). TFAP4 overexpression in WT HAP1-7TGP cells increased WNT3A-induced reporter signal by 2.6-fold but did not induce reporter activity in unstimulated cells (*Figure 2B*), suggesting TFAP4 is a limiting factor for WNT signaling in these cells. The gain- and

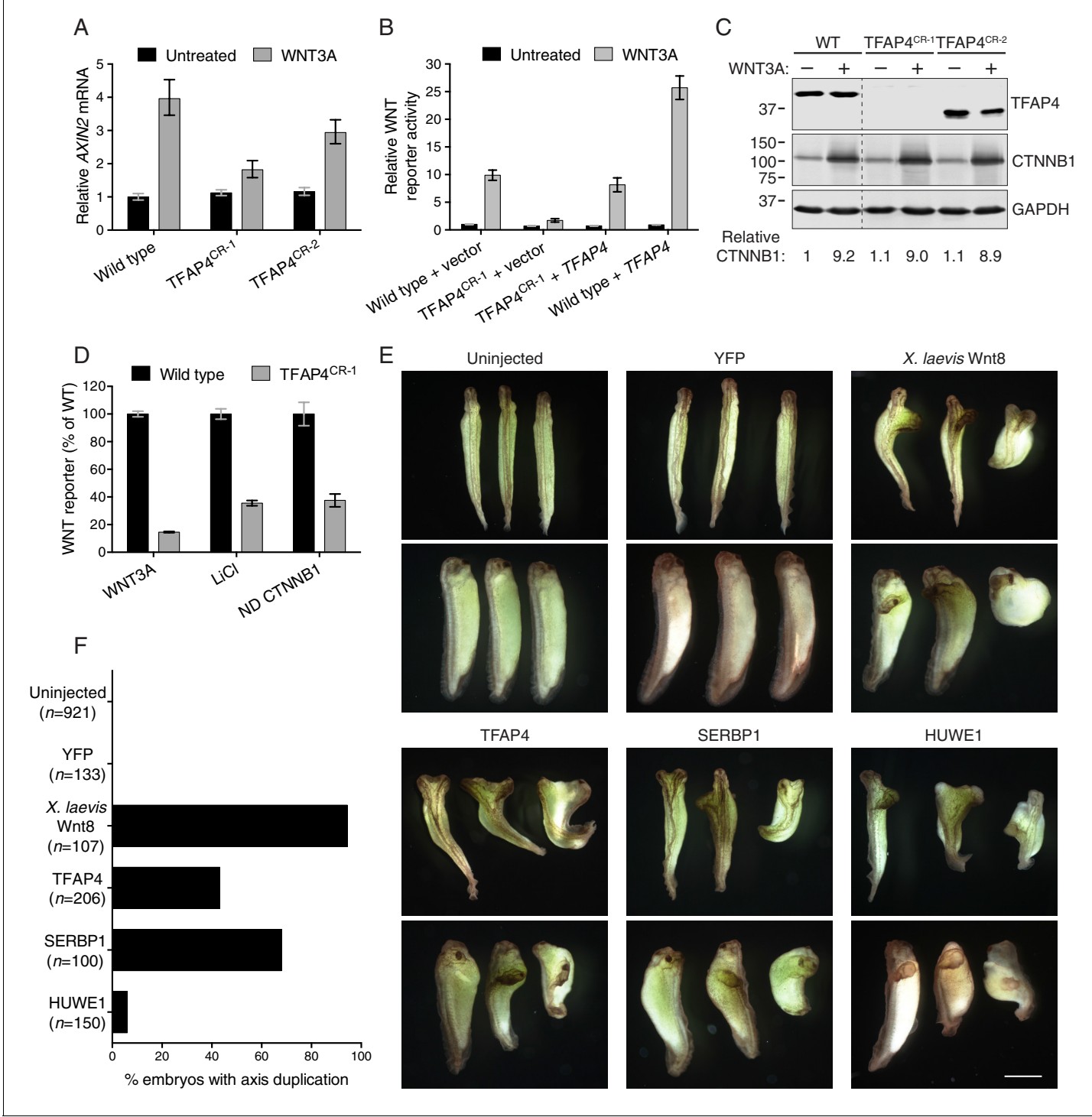

**Figure 2.** The transcription factor TFAP4 regulates WNT signaling downstream of the CTNNB1 destruction complex (A–D), and ectopic expression of TFAP4, SERBP1 and HUWE1 in *X. laevis* embryos induces secondary body axis formation (E–F). (A) *AXIN2* mRNA (average ± standard deviation (SD) *AXIN2* mRNA normalized to *HPRT1* mRNA, each measured in triplicate reactions), relative to untreated WT cells, for single WT HAP1-7TGP and TFAP4^CR clonal cell lines (see Materials and methods and *Supplementary file 2* for descriptions of all CRISPR/Cas9-engineered and GT-containing clonal cell lines). Cells were treated with 50% WNT3A CM where indicated. (B) WNT reporter activity (median ± standard error of the median (SEM) EGFP fluorescence from 1000 transfected cells), relative to untreated WT cells transfected with empty vector, for WT and TFAP4^CR-1 cells transfected with pCS2^+ empty vector or pCSDest-TFAP4 (together with pmCherry as a co-transfection marker). Cells were treated with 50% WNT3A CM where indicated. (C) Immunoblot analysis of WT and TFAP4^CR clonal cell lines treated with 50% WNT3A CM where indicated. CTNNB1 protein levels (CTNNB1 intensity normalized to GAPDH intensity), relative to untreated WT cells, are shown below the blots. Molecular weight standards (in

*Figure 2 continued on next page*

*Figure 2 continued*

kilodaltons (kDa)) are indicated on the left and the identity of the protein measured in each blot is indicated on the right. (D) WNT reporter activity (median ± SEM EGFP fluorescence from 10,000 WNT3A- or LiCl-treated cells, or from 2800 cells transfected with non-degradable (ND, S33Y mutant) CTNNB1) for WT and TFAP4^CR-1 cells, depicted as percentage of WT. Cells were treated with 50% WNT3A CM or with 40 mM of the GSK3 inhibitor LiCl, or they were transfected with ND CTNNB1 and pmCherry as a co-transfection marker. (E) Four-cell stage *X. laevis* embryos were injected ventrally with 5 ng of mRNA encoding yellow fluorescent protein (YFP), *X. laevis* Wnt8, TFAP4, SERBP1 or HUWE1 and grown to stage 34. Dorsal (top panel of each pair) and lateral (bottom panel of each pair) views for groups of three embryos are shown. Scale bar = 1 mm. (F) Percentage of embryos with a secondary body axis. The total number of injected embryos is indicated below the group name.

loss-of-function effects of TFAP4 demonstrate an important regulatory role in WNT signaling in human cells, consistent with its prominent position among the hits of the WNT screens—and indeed of several other screens described later in this work.

Because it is a transcription factor, TFAP4 is likely to function downstream of the destruction complex (*Figure 1A*). Depletion and truncation of TFAP4 in TFAP4^CR-1 and TFAP4^CR-2 cells, respectively, did not affect WNT-dependent accumulation of CTNNB1 protein, a measure of destruction complex activity (*Figure 2C*). We confirmed this conclusion by epistasis analysis, in which we activated signaling in WT HAP1-7TGP and TFAP4^CR-1 cells at various levels of the pathway by 1) addition of WNT3A, which acts at the cell surface, 2) addition of the GSK3 inhibitor LiCl, which inactivates the destruction complex, or 3) transfection with a non-degradable (ND), constitutively active CTNNB1 mutant (S33Y), which activates the transcription complex directly (*Figure 2D*). The response of TFAP4^CR-1 cells was reduced in all cases when compared to WT HAP1-7TGP cells. Thus, TFAP4 must act together with or downstream of CTNNB1.

To test if TFAP4 can influence WNT signaling during development, we employed an established body axis duplication assay in *Xenopus laevis* embryos. Activation of WNT signaling in the dorsal side of the early *X. laevis* embryo is a critical event in the formation of the Spemann organizer, an important tissue-organizing center found in vertebrates (*Spemann, 1938*), and ectopic activation of WNT signaling in the ventral side leads to formation of a second body axis. Microinjection of mRNA encoding TFAP4 into *X. laevis* embryos caused the formation of a secondary body axis (*Figure 2E and F*), demonstrating that TFAP4 can promote ectopic activation of WNT signaling during development.

Future work will focus on defining the contexts in which TFAP4 regulates WNT transcriptional responses under physiological and pathological conditions, given that its site of action downstream of the CTNNB1 destruction complex could be favorable for therapeutic intervention in cancers where WNT signaling is activated by loss of APC or by mutations that stabilize CTNNB1.

## The C-terminal DAX domain of AXIN2 controls CTNNB1 transcriptional activity

*AXIN* genes encode the principal scaffold (reviewed in *Tacchelly-Benites et al., 2013*; *Song et al., 2014*) and limiting component (*Lee et al., 2003*) of the CTNNB1 destruction complex. The two paralogues in mammals, *AXIN1* and *AXIN2*, are functionally redundant (*Chia and Costantini, 2005*), although their expression patterns are quite distinct: *AXIN1* is expressed ubiquitously, while *AXIN2* is expressed at low levels in the absence of WNT signals (*Jho et al., 2002*). *AXIN2* is also the key component of a negative feedback loop in the WNT pathway (*Lustig et al., 2002*). As a universal and direct target gene of the pathway, its increased expression following stimulation with WNT can lead to elevated levels of the destruction complex and, consequently, reduced levels of CTNNB1. Given this well-established negative regulatory role, the enrichment of GT insertions mapping to *AXIN2* in HAP1-7TGP cells with *reduced* WNT reporter activity recovered during the WNT screens (*Figure 1E and F*, and *Supplementary file 1*) presented us with a paradox.

An important clue emerged from a careful inspection of the distribution of GT insertions mapping to *AXIN2* in the sorted cells. In most hits from haploid genetic screens, exemplified by *LRP6* (*Figure 3A*), GT insertions cluster at the 5' end of the gene because of the propensity of retroviral integration near transcriptional start sites and because such insertions are likely to generate null alleles (*Carette et al., 2011a*). Contrary to this general case, nearly all GT insertions in *AXIN2* mapped to the opposite end of the gene in the last intron (*Figure 3A*). These insertions are

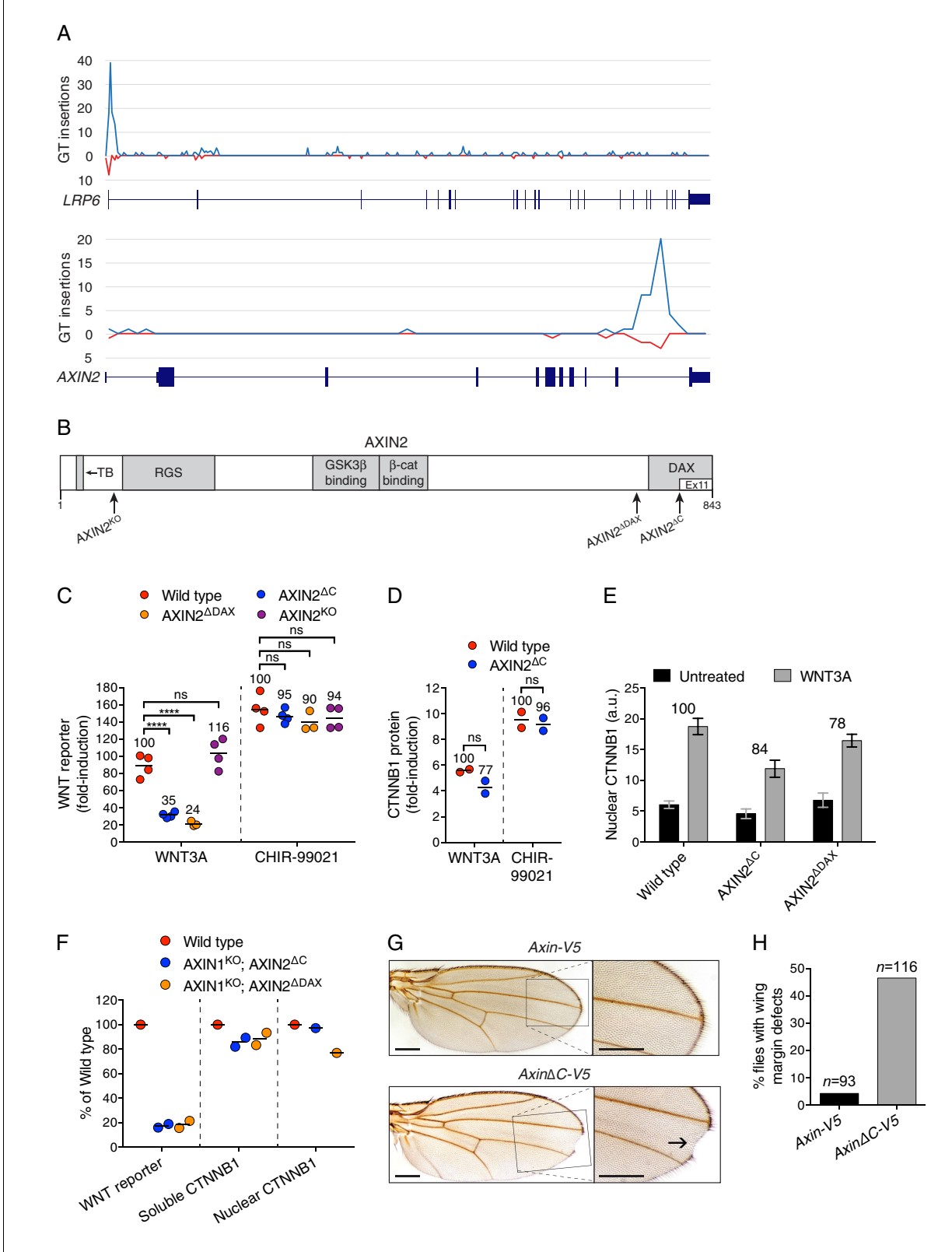

**Figure 3.** The C-terminal DAX domain of AXIN2 controls CTNNB1 transcriptional activity. (**A**) GT insertions in *LRP6* (top histogram) and *AXIN2* (bottom histogram) mapped for the sorted cells from the WNT positive regulator, low stringency screen (*Figure 1E*). The histograms depict the number of GT integrations in the sense (blue) or antisense (red) orientation, relative to the coding sequence of the gene, within consecutive 500 base pair (bp) intervals along the length of each gene. Due to the directionality of the splice acceptor in the GT, typically only sense GT insertions in introns disrupt

*Figure 3 continued on next page*

*Figure 3 continued*

the gene, whereas GT insertions in exons generally disrupt the gene regardless of orientation. RefSeq gene tracks for *LRP6* and *AXIN2* are shown beneath each histogram following the University of California, Santa Cruz (UCSC) genome browser display conventions: coding exons are represented by thick blocks, UTRs by thin blocks, and introns by horizontal lines connecting the blocks. Both genes are displayed with their 5' ends to the left, and encompass chromosome 12, bps 12267499–12116000 for *LRP6*, and chromosome 17, bps 65561999–65528500 for *AXIN2* (hg18). (B) Schematic representation of the human AXIN2 protein drawn to scale in the horizontal dimension. Amino acid numbers are indicated below, and arrows show the sites at which truncations were made by CRISPR/Cas9-mediated genome editing in the indicated cell lines. Known domains, regions and motifs (based on UniProt annotation) are depicted in gray (TB, tankyrase-binding motif). Exon 11, eliminated by GT insertions found in cells sorted during the WNT screens for positive regulators, is delineated by a thinner white block. (C) Fold-induction in WNT reporter (median EGFP fluorescence from 20,000 cells) following treatment with 50% WNT3A CM or 10 µM of the GSK3 inhibitor CHIR-99021. Each circle represents a unique clonal cell line (determined by genotyping, *Supplementary file 2*), and the average of three to four independent clones for each genotype is indicated by a horizontal line. For each treatment, percentage reporter activation relative to WT cells is also indicated above each group of circles to facilitate comparisons. Significance was determined by one-way ANOVA, and is indicated as ****($p$<0.0001) or ns (not significant). (D) Fold-induction in soluble CTNNB1 protein (average CTNNB1 intensity normalized to ACTIN intensity from duplicate immunoblots) following treatment with 50% WNT3A CM or 10 µM CHIR-99021. Each circle represents a unique clonal cell line, and the average of two independent clones for each genotype is indicated by a horizontal line. Significance was determined by unpaired t-test with Welch's correction. Representative immunoblots used for quantification of CTNNB1 and ACTIN are shown in *Figure 3—figure supplement 1C*. (E) Nuclear CTNNB1 (average nuclear fluorescence per unit area from three fields of view) in single clonal cell lines of the indicated genotypes was quantified as described in Materials and methods. Cells were treated with 50% WNT3A CM where indicated. For each cell line, the fold-increase in CTNNB1 nuclear accumulation following treatment with WNT3A, expressed as percentage of WT, is also indicated above the bars to facilitate comparisons. For WNT3A-treated cells, differences in nuclear CTNNB1 between WT and mutant cells were not statistically significant as determined by one-way ANOVA. Examples of confocal sections used for quantification of nuclear CTNNB1 are shown in *Figure 3—figure supplement 2B–2D*. (F) WNT reporter activity (median EGFP fluorescence from 10,000 cells), soluble CTNNB1 protein (average CTNNB1 intensity normalized to ACTIN intensity from duplicate immunoblots), and nuclear CTNNB1 protein (average nuclear fluorescence per unit area from 2 to 3 fields of view), depicted as percentage of WT, for cells treated with 50% WNT3A CM. Each circle represents a unique clonal cell line, and where applicable the average of two independent clones is indicated by a horizontal line. (G) Adult *D. melanogaster* wings expressing *Axin-V5* (top images) or *AxinΔC-V5* (bottom images) under the control of the c765-Gal4 driver. Loss of sensory bristles and tissue at the wing margin, indicative of impaired Wg signaling, is shown (arrow) in the higher magnification view of the delineated area. While loss of Notch signaling can also result in wing margin defects due to a requirement of Notch signaling for Wg expression at the dorso-ventral boundary in the wing imaginal disc (*Diaz-Benjumea and Cohen, 1995*), we ruled out that possibility by confirming intact Wg expression in the wing imaginal disc of flies expressing *AxinΔC-V5* (*Figure 3—figure supplement 3B*). Scale bars = 20 µm. (H) Percentage of flies with wing margin defects. 4.3% of flies expressing *Axin-V5* exhibited loss of bristles at the wing margin, but no loss of wing tissue; 46.6% of flies expressing *AxinΔC-V5* exhibited loss of bristles at the wing margin as well as loss of wing tissue.

The following figure supplements are available for figure 3:

**Figure supplement 1.** AXIN1 and AXIN2 are redundant in haploid human cells (A–B), and CTNNB1 protein is stabilized normally in AXIN2$^{ΔC}$ cells (C).

**Figure supplement 2.** Cells lacking the C-terminal DAX domain of AXIN2 exhibit normal CTNNB1 nuclear accumulation following WNT treatment.

**Figure supplement 3.** Expression of an *Axin* transgene encoding C-terminally truncated protein compromises Wg signaling in *D. melanogaster*.

predicted to produce a truncated AXIN2 protein product lacking exon 11 (*Figure 3B*), comprising half of the DAX domain, which has been implicated both in CTNNB1 destruction complex function and in interactions with the receptor complex at the plasma membrane (reviewed in *Tacchelly-Benites et al., 2013*; *Song et al., 2014*).

The results of the WNT screens (*Figure 1E and F*, and *Supplementary file 1*) suggested that deletion of half of the DAX domain of AXIN2 reduces responsiveness to WNT. The ease of genome editing in haploid cells enabled us to precisely assess the magnitude of signaling defects by comparing multiple independent WT and mutant HAP1-7TGP clonal cell lines containing specific mutations in the single endogenous *AXIN2* allele, which minimized the likelihood of non-specific effects and accounted for interclonal variability. We used CRISPR/Cas9 to generate HAP1-7TGP cell lines lacking exon 11 of *AXIN2* (designated AXIN2$^{ΔC}$, *Figure 3B* and *Supplementary file 2*). Since incomplete protein domains can remain unfolded, we also generated multiple cell lines lacking the complete DAX domain (AXIN2$^{ΔDAX}$) and, as a control, cell lines lacking *AXIN2* entirely (AXIN2$^{KO}$) (*Figure 3B* and *Supplementary file 2*). AXIN2$^{KO}$ clones exhibited no defect in WNT3A-dependent reporter induction (*Figure 3C*) due to redundancy with AXIN1 (*Figure 3—figure supplement 1A and B*, and *Table 1*). However, AXIN2$^{ΔC}$ and AXIN2$^{ΔDAX}$ clones exhibited, on average, a 65% and a 76% reduction in WNT3A-induced signaling, respectively, compared to WT HAP1-7TGP cells (*Figure 3C*). All

AXIN2 mutant cell lines responded normally to the GSK3 inhibitor CHIR-99021 (*Figure 3C*), demonstrating that the reduction in WNT3A-induced signaling was not due to defects in downstream steps or to irrelevant factors affecting reporter fluorescence. The fact that C-terminally truncated AXIN2 reduced signaling induced by WNT3A explained why GT insertions generating this unusual *AXIN2* allele were enriched in the WNT screens for positive regulators. Since HAP1-7TGP cells also express AXIN1 (*Figure 3—figure supplement 1A* and *Table 1*), the effect of C-terminally truncated AXIN2 must be dominant.

We asked whether the reduction in WNT signaling caused by truncated AXIN2 was due to a defect in WNT3A-induced CTNNB1 stabilization. Surprisingly, WNT3A robustly promoted the stabilization of soluble CTNNB1 protein in AXIN2$^{\Delta C}$ cells (*Figure 3D* and *Figure 3—figure supplement 1C*), revealing a disconnect between CTNNB1 protein abundance and transcriptional activity. CTNNB1 accumulation in response to CHIR-99021 was also normal in AXIN2$^{\Delta C}$ cells (*Figure 3D*, and *Figure 3—figure supplement 1C*). The discrepancy between CTNNB1 stability and transcriptional activity was not caused by defective nuclear accumulation; in both AXIN2$^{\Delta C}$ and AXIN2$^{\Delta DAX}$ cells, accumulation of nuclear CTNNB1 following WNT treatment was largely normal (*Figure 3E* and *Figure 3—figure supplement 2*). We conclude from these experiments that deleting the DAX domain of AXIN2 dissociates CTNNB1 protein abundance from its transcriptional activity.

These effects could reflect an autonomous property of the AXIN2 protein lacking the DAX domain, or a more complex interaction with the remaining WT AXIN1. To rule out confounding effects due to AXIN1, we disrupted the single allele of *AXIN1* in individual AXIN2$^{\Delta C}$ and AXIN2$^{\Delta DAX}$ clonal cell lines to generate double-mutant cell lines (AXIN1$^{KO}$; AXIN2$^{\Delta C}$ and AXIN1$^{KO}$; AXIN2$^{\Delta DAX}$, respectively, *Supplementary file 2* and *Figure 3—figure supplement 1A*). The only AXIN protein present in these cells is truncated AXIN2 lacking either half or the entire DAX domain. Truncated AXIN2 caused the same effects upon elimination of AXIN1—decreased WNT3A-induced reporter activity despite normal accumulation of soluble and nuclear CTNNB1 (*Figure 3F* and *Figure 3—figure supplement 2*). These findings provide further evidence that AXIN2 truncations disrupt CTNNB1-mediated transcription through a novel mechanism intrinsic to this allele.

We tested the generality of our findings by introducing *Axin* transgenes into the fly *Drosophila melanogaster*, a model organism that has been used extensively for genetic studies of WNT signaling. We generated a transgene encoding an epitope-tagged (V5) fusion of the single *D. melanogaster* Axin protein lacking the last 41 amino acids (Axin$\Delta$C-V5, *Figure 3—figure supplement 3A*). These amino acids correspond to those encoded by exon 11 of human *AXIN2*, the exon disrupted by GT insertions in the WNT screens (*Figure 3A and B*). Expression of *Axin$\Delta$C-V5* impaired Wingless (Wg, fly WNT) signaling based on target gene expression and phenotypic readouts during both embryonic and larval development (*Figure 3G and H*, and *Figure 3—figure supplement 3B–H*). The observed defects in Wg signaling were not due to decreased expression of Wg itself (*Figure 3—figure supplement 3B and C*) or to increased expression of Axin$\Delta$C-V5 protein (*Figure 3—figure supplement 3E and F*). In control experiments, expression of WT *Axin-V5* (*Figure 3—figure supplement 3A*) at physiological levels (*Wang et al., 2016*) using the same promoter as for *Axin$\Delta$C-V5* did not disrupt Wg signaling (*Figure 3G and H*, and *Figure 3—figure supplement 3B–H*). The fact that in flies, like in haploid human cells, expression of an *Axin* transgene lacking the C-terminal domain reduces Wg signaling even in the presence of endogenous *Axin* is consistent with a dominant effect that restrains CTNNB1 transcriptional activity.

In summary, unbiased genome-wide screens for positive regulators of WNT signaling uncovered an unsuspected role for the C-terminal DAX domain of AXIN2 in controlling CTNNB1 transcriptional activity, since deletion of this domain led to a severely compromised transcriptional response despite normal accumulation of CTNNB1 protein. While the mechanistic basis of this process remains to be elucidated, our results cannot be explained by previously described functions of the AXIN DAX domain. The DAX domain has been implicated in AXIN polymerization, in interactions with DVL, and in mediating an intramolecular, auto-inhibitory interaction that allows the receptor complex to inactivate the destruction complex in a catalytic manner (*Fiedler et al., 2011*; *Kim et al., 2013*). These models predict that loss of the DAX domain would impair communication between the receptor complex and the destruction complex, leading to defective WNT-induced CTNNB1 stabilization, in contrast to our results (*Figure 3D and F*). Thus, the disconnect between CTNNB1 abundance and transcriptional activity caused by deletion of the DAX domain demonstrates a new biochemical function for this domain.

The discovery of this dominant allele of *AXIN2* was made possible by the untargeted nature of GT-based insertional mutagenesis and thus would not have emerged from strict loss-of-function screens such as those mediated by RNA interference or CRISPR/Cas9. Given that rare dominant alleles can provide mechanistic insights distinct from those afforded by null alleles, our findings justify the design of comprehensive 'exome-wide' sgRNA libraries for CRISPR/Cas9-based screens.

## A comparative analysis of screens uncovers requirements for RSPO-potentiated signaling in response to low levels of WNT

RSPOs are stem cell growth factors that potentiate responses to WNT ligands by binding to LGR-family receptors and neutralizing the ZNRF3 and RNF43 E3 ubiquitin ligases to increase levels of WNT receptors on the cell surface (reviewed in *de Lau et al., 2014*; *Figure 1A*). Recurrent translocations in genes encoding RSPOs are found in some colorectal tumors and targeting the resulting fusion proteins blocks tumorigenesis (*Seshagiri et al., 2012*; *Storm et al., 2016*). Mutations in RNF43 that mimic the effect of stimulation with RSPO have also been reported in multiple cancers (*Giannakis et al., 2014*). Regulators of RSPO-enhanced WNT signaling could therefore be important both in normal physiological and in pathological contexts.

HAP1-7TGP cells were responsive to RSPO-mediated effects on WNT signaling. RSPO1 markedly amplified the reporter response to low concentrations of WNT3A CM but was completely inactive in the absence of WNT (*Figure 4—figure supplement 1A*). We determined the concentration of WNT3A CM at which responsiveness to RSPO1 was maximal (*Figure 4—figure supplement 1B*) and used these conditions in a sensitized genome-wide screen for mediators of RSPO-enhanced WNT signaling. Notably, the concentration of WNT used in this screen, henceforth referred to as the 'low WNT + RSPO screen,' was 49-fold lower than that used in the WNT screens for positive regulators (*Figure 1E and F*).

Following treatment with WNT3A CM plus RSPO1, we isolated cells with the lowest 7% EGFP fluorescence ('lowest 7%' gate in *Figure 4—figure supplement 1C*). After four consecutive sorts, which resulted in a marked enrichment of cells with diminished responsiveness to WNT3A CM plus RSPO1 (*Figure 4—figure supplement 1C*), we sequenced and mapped GT integrations (*Figure 4A* and *Supplementary file 1*). Reassuringly, the top hit was *LGR4*, the gene encoding the RSPO1 receptor, confirming that the screen was sensitive to requirements for RSPO1-dependent signaling. Top hits of this screen included many genes encoding known WNT regulators also uncovered in the WNT screens for positive regulators (*Figure 1E and F*, and *Supplementary file 1*): *LRP6*, *AXIN2*, *DOT1L*, *MLL*, *CTNNB1*, *CREBBP*, *BCL9* and *RNF146*. *TFAP4*, encoding the required transcription factor we described earlier in this study, was also among the top hits (*Figure 4A* and *Supplementary file 1*).

We distinguished genes selectively required for RSPO-enhanced WNT signaling through a comparative analysis of the low WNT + RSPO screen (*Figure 4A* and *Supplementary file 1*) and both WNT screens for positive regulators conducted at a near-saturating dose of WNT (*Figure 1E and F*, and *Supplementary file 1*). For this analysis, we used two different measures of gene disruption caused by GT integrations (see Materials and methods, *Figure 4B* and *Supplementary file 3*). First, the FDR-corrected $p$-value, reflecting enrichment of GT integrations in the sorted vs. the unsorted cells from each screen, was used to set a stringent cutoff for inclusion of hits in the analysis. Second, an Intronic GT Insertion Orientation Bias (IGTIOB) score, reflecting enrichment of sense vs. antisense GT integrations (relative to the coding sequence of the gene) in introns only for the sorted cells from each screen, was used to compare hits between screens. The IGTIOB score relies on the fact that generally only sense GT insertions in introns should inactivate genes due to the directionality of the splice acceptor. Genes selectively required for RSPO-enhanced WNT signaling should show a pattern of GT enrichment similar to that of the gene encoding the RSPO receptor, *LGR4*. Conversely, regulators required for WNT signaling under all treatment conditions should be equally enriched for GT integrations in all screens (*Figure 4B* and *Supplementary file 3*).

The most striking outcome of this analysis was the identification of multiple genes encoding components of the GPI-anchor biosynthetic pathway that were enriched for GT insertions in the low WNT + RSPO screen but not the WNT screens (*Figure 4B* and *Supplementary file 3*). Fourteen genes in the GPI biosynthesis pathway had an FDR-corrected $p$-value<0.05 (*Supplementary file 1* and *Figure 4—figure supplement 1D*). Therefore, a GPI-anchored protein may be particularly important in mediating signaling triggered by a combination of RSPO and a low dose of WNT. The

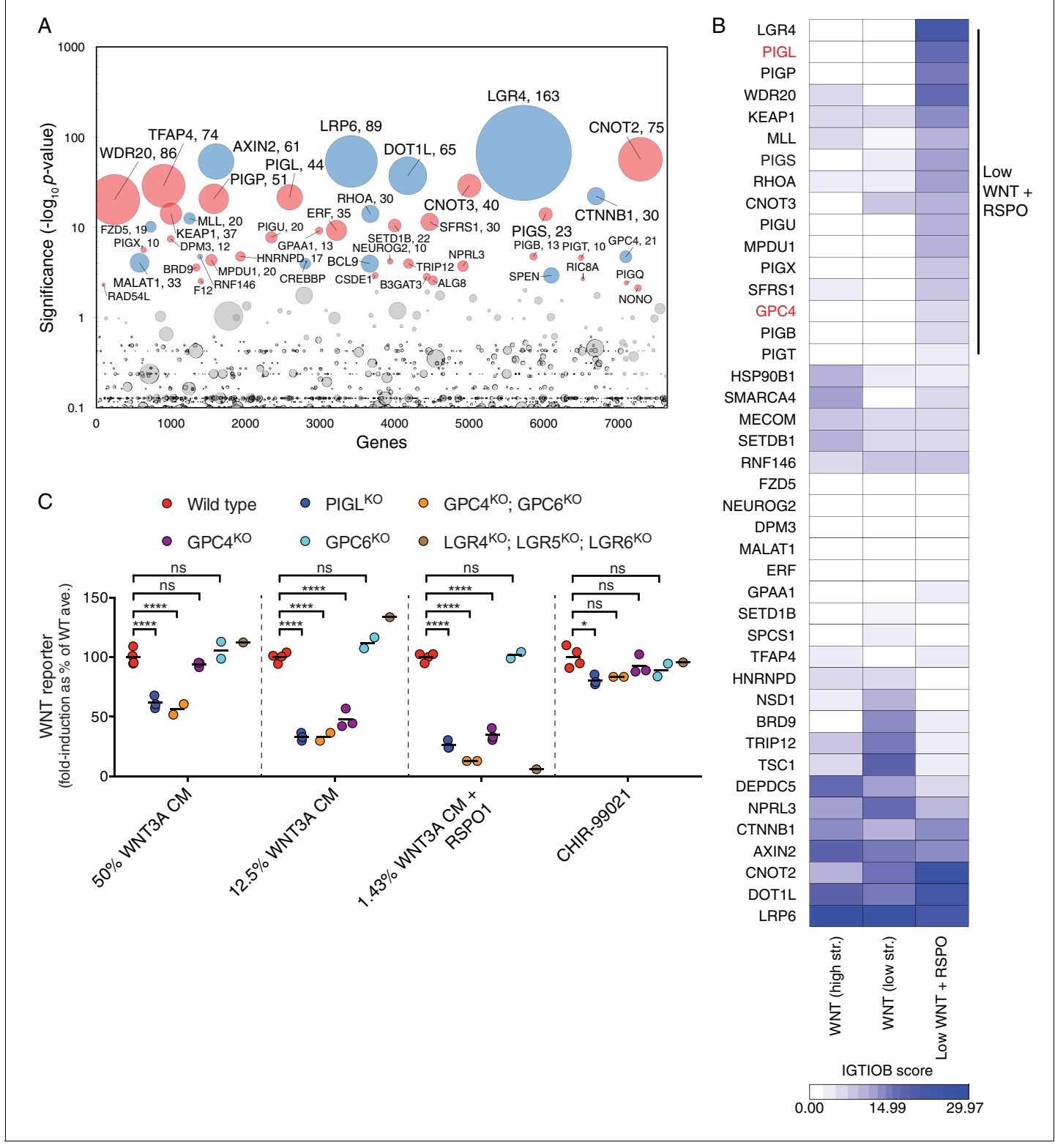

**Figure 4.** A comparative analysis of screens uncovers requirements for RSPO-potentiated signaling in response to low levels of WNT. (**A**) Circle plot depicting genes enriched for GT insertions in the low WNT + RSPO screen for regulators of RSPO-enhanced WNT signaling. See legend to *Figure 1C–F* for details. (**B**) Heat map comparing the two WNT positive regulator screens (*Figure 1E and F*) and the low WNT + RSPO screen (*Figure 4A*). Genes enriched for GT insertions (FDR-corrected *p*-value<$10^{-4}$) in at least one of the three screens were clustered based on their IGTIOB score in each screen (see Materials and methods and *Supplementary file 3*). A group of genes preferentially enriched for GT insertions in the low WNT + RSPO screen is

*Figure 4 continued on next page*

Figure 4 continued

indicated, headlined by the RSPO receptor LGR4. Genes selected for a detailed analysis are labeled in red. (C) Fold-induction in WNT reporter (average EGFP fluorescence from 10,000 cells) following treatment with 50% WNT3A CM, 12.5% WNT3A CM, 1.43% WNT3A CM + 20 ng/ml RSPO1 or 10 μM of the GSK3 inhibitor CHIR-99021, expressed as percentage of the average for WT cells to facilitate comparisons. Each circle represents the fold-induction for a unique clonal cell line (determined by genotyping, *Supplementary file 2*), and where applicable, the average of two to four independent clones for each genotype is indicated by a horizontal line. Significance was determined by one-way ANOVA, and is indicated as **** (p<0.0001), *(p<0.05) or ns (not significant).

The following figure supplement is available for figure 4:

**Figure supplement 1.** Comparative genetic screens uncover requirements for RSPO-potentiated signaling in response to low levels of WNT.

gene encoding the glypican GPC4, a GPI-linked heparan sulfate proteoglycan (HSPG), was also preferentially enriched for GT insertions in the low WNT + RSPO screen (*Figure 4B* and *Supplementary file 3*), with an FDR-corrected *p*-value=$1.92 \times 10^{-5}$, more significant than those of established WNT signaling components such as *CREBBP* and *BCL9* (*Figure 4A* and *Supplementary file 1*). Glypicans are important for concentrating extracellular ligands at the cell surface, and GPC4 has been proposed to bind and concentrate WNT3A and WNT5A in the vicinity of their receptors (*Sakane et al., 2012*). However, since neither GPI biosynthesis nor glypican genes were significant hits in the WNT screens for positive regulators (*Figure 1E and F*, and *Supplementary file 1*), we hypothesized that these genes must play a crucial role under the conditions of the low WNT + RSPO screen, either by mediating RSPO responsiveness like *LGR4*, or by selectively affecting reception of WNT3A at the very low concentration used in this screen.

To distinguish between these two possibilities, we analyzed the signaling response to WNT3A alone or to a low concentration of WNT3A plus RSPO1 in clonal HAP1-7TGP cell lines in which we disrupted *PIGL*, a gene in the GPI biosynthesis pathway (*Figure 4—figure supplement 1D*), or *GPC4* (designated PIGL$^{KO}$ and GPC4$^{KO}$, respectively, *Supplementary file 2*). The glypican GPC6 is redundant with GPC4 in certain contexts (*Allen et al., 2012*), so we also generated HAP1-7TGP cell lines in which we disrupted *GPC6* alone or in combination with *GPC4* (designated GPC6$^{KO}$ and GPC4$^{KO}$; GPC6$^{KO}$, respectively, *Supplementary file 2*). As a control we generated a HAP1-7TGP cell line lacking all three RSPO receptors (designated LGR4$^{KO}$; LGR5$^{KO}$; LGR6$^{KO}$, *Supplementary file 2*).

As expected from the role of LGRs as exclusive mediators of responsiveness to RSPO but not to WNT, LGR4$^{KO}$; LGR5$^{KO}$; LGR6$^{KO}$ cells did not respond to RSPO1 in the presence of a low dose of WNT3A, but exhibited no signaling defects when stimulated with higher doses of WNT3A alone (*Figure 4C* and *Figure 4—figure supplement 1E*). In contrast, PIGL$^{KO}$ and GPC4$^{KO}$; GPC6$^{KO}$ cells manifested some reduction in signaling when stimulated with a near-saturating dose of WNT3A alone, but this reduction was more pronounced following treatment with lower doses of WNT3A alone or a low dose of WNT3A combined with RSPO1 (*Figure 4C* and *Figure 4—figure supplement 1E*). GPC4$^{KO}$ cells were stimulated normally by a near-saturating dose of WNT3A and exhibited a smaller defect than GPC4$^{KO}$; GPC6$^{KO}$ when stimulated with a lower dose of WNT3A alone or in combination with RSPO1, while GPC6$^{KO}$ cells had no signaling defect at all (*Figure 4C*). These results suggest that GPC4 and GPC6 are partially redundant in HAP1 cells, since they are both expressed albeit at very different levels (*Table 1*). In a control experiment, WNT signaling induced by the GSK3 inhibitor CHIR-99021 was largely unaffected in all mutant cell lines (*Figure 4C*), demonstrating that there were no signaling defects downstream of the receptor complex (*Figure 1A*).

Taken together these results indicate that genes in the GPI biosynthesis pathway and *GPC4/6* are required for signaling in response to low levels of WNT, and explain why they may have been more prominent hits in the low WNT + RSPO than in the WNT screens (*Figures 1E,F* and *4A* and *Supplementary file 1*). Presently, we cannot confirm or discount an additional, direct contribution of GPC4/6 or another GPI-anchored protein to RSPO reception, as we have been unable to directly measure responses to RSPO alone in HAP1 cells. Yet, the presence in all RSPOs of a thrombospondin domain capable of binding heparin sulfate and mediating interactions with HSPGs such as glypicans (*Nam et al., 2006*; *Ohkawara et al., 2011*) makes this an intriguing possibility.

In summary, our comparative analysis shows that forward genetic screens in haploid human cells are exquisitely sensitive to both the identity and concentration of ligands used to initiate signaling. They can uncover ligand-specific receptors, such as LGR4, or accessory factors that are rate-limiting for signaling only under specific regimes of ligand concentrations, such as GPI biosynthetic enzymes and glypicans. Of note, the low WNT + RSPO screen (*Figure 4A* and *Supplementary file 1*) was sensitive enough to reveal redundant regulators, such as *FZD5* and *DVL3* (*Table 1*), that were not significant hits under the near-saturating WNT3A dose used in the WNT screens (*Figure 1E and F*, and *Supplementary file 1*).

## Suppressor screens reveal genotype-specific signaling requirements

Given the potential of comparative screens to identify context-specific regulators, we searched for genes whose inactivation would suppress the pathological signaling that ensues when key negative regulators of the WNT pathway are lost. Since negative regulators such as APC are frequently mutated in cancer, suppressor mutations and the mechanisms through which the affected genes regulate signaling may reveal therapeutic targets.

Our initial screen for rate-limiting negative regulators of WNT signaling (*Figure 1C* and *Supplementary file 1*) suggested that disruption of the single allele of *APC* or *CSNK1A1* in HAP1-7TGP cells should lead to constitutive activation of the pathway. We designed two screens to uncover suppressors of ligand-independent signaling induced by loss of *APC* or *CSNK1A1* (*Figure 5A*). We disrupted *APC* or *CSNK1A1* in HAP1-7TGP cells using CRISPR/Cas9 and isolated two clonal cell lines designated APC$^{KO-1}$ and CSNK1A1$^{KO-1}$, respectively (*Supplementary file 2*). Sequencing revealed frameshift mutations in the single allele of each gene. We confirmed by immunoblotting that the APC signal was reduced by >96.5% in APC$^{KO-1}$ cells, and CSNK1A1 was undetectable in CSNK1A1$^{KO-1}$ cells (*Figure 5—figure supplement 1A*). The level of constitutive WNT reporter fluorescence in both the APC$^{KO-1}$ and CSNK1A1$^{KO-1}$ clones was higher than that induced by near-saturating WNT3A or by the GSK3 inhibitor CHIR-99021 in WT HAP1-7TGP cells (*Figure 5—figure supplement 1B*).

In two independent screens, henceforth referred to as the 'APC suppressor screen' and the 'CSNK1A1 suppressor screen,' respectively, we mutagenized APC$^{KO-1}$ and CSNK1A1$^{KO-1}$ cells with GT retrovirus and enriched for cells with the lowest 10% WNT reporter fluorescence ('lowest 10%' gate in *Figure 5—figure supplement 1C and D*). Following two sequential rounds of sorting and amplification, 43% of APC$^{KO-1}$ and 42% of CSNK1A1$^{KO-1}$ cells were within this gate (*Figure 5—figure supplement 1C and D*). We analyzed the cells sorted during each screen and their respective unsorted control populations for enrichment of GT insertions (*Figure 5B and C*, and *Supplementary file 1*).

A three-way comparative analysis of the WNT positive regulator, low stringency screen in WT HAP1-7TGP cells (*Figure 1E*), the APC suppressor screen and the CSNK1A1 suppressor screen (all done at similar selection stringencies) revealed expected similarities and differences based on established epistatic relationships, as well as a number of unexpected findings (*Figure 5D* and *Supplementary file 3*). The isogenic background of the cells in these three screens and the very high statistical significance of the top hits (only genes with an FDR-corrected $p$-value$<10^{-4}$ in at least one screen were included in this analysis) enabled us to make meaningful predictions, some of which we confirmed experimentally.

Several groups of genes were clearly discernible based on their GT insertion enrichment patterns across the three screens (*Figure 5D* and *Supplementary file 3*). As expected, genes encoding components of the pathway that function downstream of the destruction complex, including *CTNNB1* and *CREBBP*, were enriched for GT insertions in all three screens (*Figure 5D* and *Supplementary file 3*). *TFAP4* was also significantly enriched for GT insertions in all three screens (*Supplementary file 3*), and it acts downstream of the destruction complex, as we confirmed experimentally (*Figure 2*). However, *TFAP4* had a low IGTIOB score in all screens (*Figure 5D* and *Supplementary file 3*) because it represents a rare case of a gene that can be disrupted by both sense and antisense GT insertions in an intron (*Supplementary file 1*), a finding that will be described in detail elsewhere.

Also as expected, genes encoding components of the pathway upstream of the destruction complex, such as *LRP6*, were predominantly enriched for GT insertions in the WNT, but not the APC or CSNK1A1 suppressor screens (*Figure 5D* and *Supplementary file 3*). *AXIN2* was also enriched for

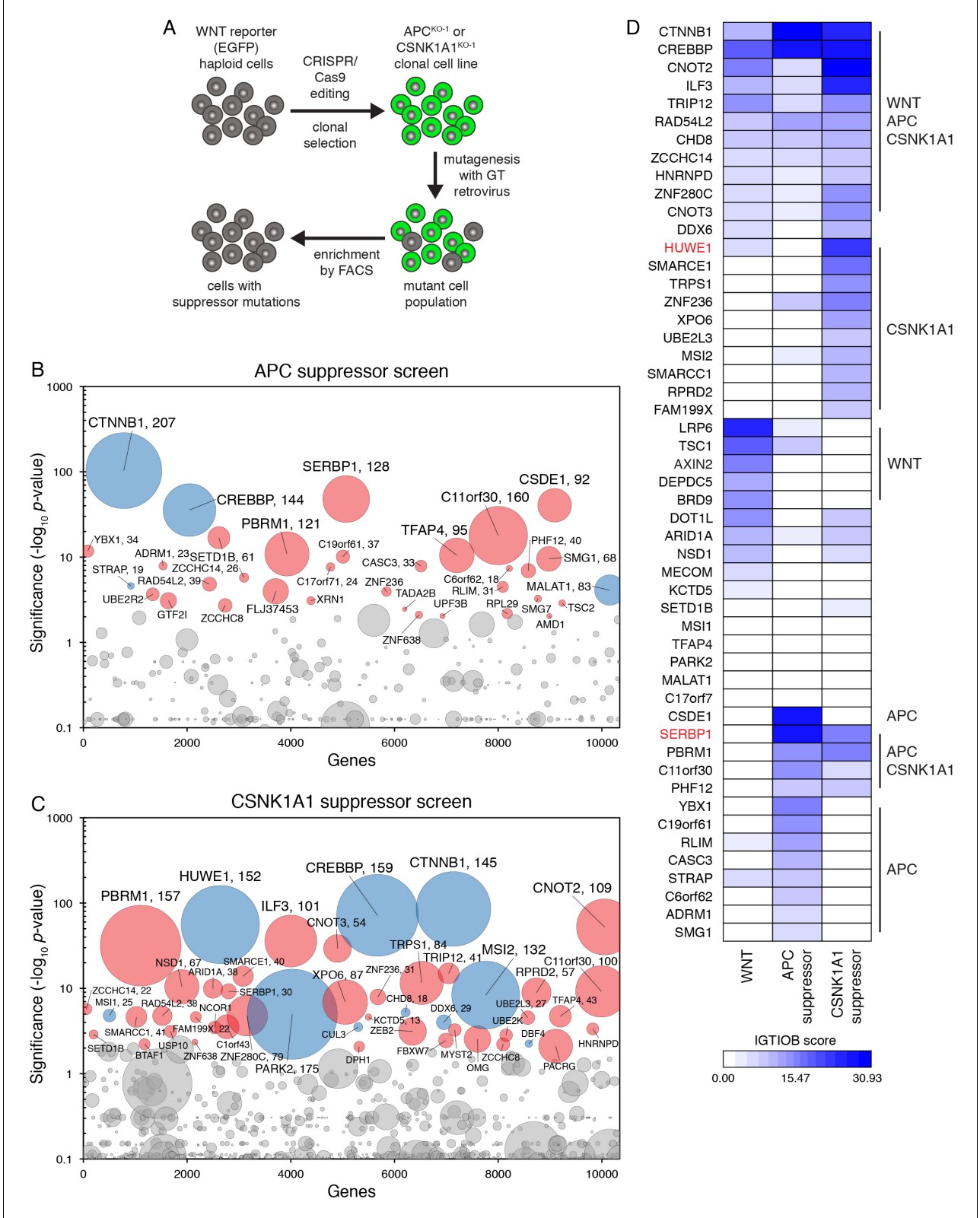

**Figure 5.** Suppressor screens in cells lacking APC or CSNK1A1 reveal genotype-specific signaling requirements. (**A**) Schematic of WNT reporter-based suppressor screens. *APC* or *CSNK1A1* was disrupted by CRISPR/Cas9-mediated genome editing of the WNT reporter haploid cell line HAP1-7TGP. Individual clonal cell lines were isolated (APC^KO-1 and CSNK1A1^KO-1, ***Supplementary file 2***) and mutagenized using GT retrovirus. Cells with reduced reporter activity were enriched by FACS to identify suppressor mutations. (**B–C**) Circle plots depicting genes enriched for GT insertions in suppressor

*Figure 5 continued on next page*

*Figure 5 continued*

screens in which constitutive WNT signaling was induced by loss of APC (B) or CSNK1A1 (C). See legend to *Figure 1C–1F* for details. (D) Heat map comparing the WNT positive regulator, low stringency screen (*Figure 1E*), and the APC and CSNK1A1 suppressor screens (*Figure 5B and C*). Genes enriched for GT insertions (FDR-corrected $p$-value$<10^{-4}$ ) in at least one of the three screens were clustered based on their IGTIOB score in each screen (see Materials and methods and *Supplementary file 3*). Classes of genes preferentially enriched for GT insertions in various screens are indicated. Genes selected for a detailed analysis are labeled in red.

The following figure supplement is available for figure 5:

**Figure supplement 1.** Suppressor screens in cells lacking APC or CSNK1A1 reveal genotype-specific signaling requirements.

GT insertions in the WNT screen exclusively, suggesting that the mechanism responsible for reducing WNT responsiveness in cells containing AXIN2 truncations depends on other components of the CTNNB1 destruction complex.

Hits enriched for GT insertions in both the APC and CSNK1A1 supprssor screens, but not the WNT screen, likely represent a class of genes capable of modulating WNT signaling in the absence of destruction complex activity. The most prominent hit in this category was *SERBP1* (*Figure 5D* and *Supplementary file 3*), encoding an RNA binding protein that has not been previously implicated in WNT signaling. In the section that follows we explored how SERBP1 regulates WNT signaling in cells lacking destruction complex activity.

Genes enriched for GT insertions predominantly in the APC suppressor screen included those encoding various RNA binding proteins, components of the mRNA nonsense-mediated decay pathway, and negative regulators of RNA polymerase (*Figure 5D* and *Supplementary file 3*), suggesting a connection between RNA metabolism and signaling in APC$^{KO-1}$ cells.

Surprisingly, there were a number of genes predominantly enriched for GT insertions in the CSNK1A1 suppressor screen (*Figure 5D* and *Supplementary file 3*). Given that the principal role of CSNK1A1 in WNT signaling is thought to be phosphorylation of CTNNB1 through the destruction complex, it was not obvious why these same genes were not enriched for GT insertions in the APC suppressor screen, where destruction complex activity was also disabled. The existence of this class of genes, apparently required for WNT signaling only in cells lacking CSNK1A1, suggested a role for CSNK1A1 in WNT signaling independent of the destruction complex. The most prominent gene in this class encodes HUWE1, an E3 ubiquitin ligase that has been proposed to downregulate WNT signaling by ubiquitinating DVL and preventing its multimerization (*de Groot et al., 2014*). In contrast, the fact that mutations of HUWE1 caused a reduction in WNT reporter fluorescence during the CSNK1A1 suppressor screen suggests a positive regulatory role. Below, we describe the peculiar role of HUWE1 in mediating WNT signaling specifically in the context of CSNK1A1 loss.

*APC* is a prototypical human tumor suppressor gene frequently lost in both sporadic and familial CRC. Importantly, reduction of WNT signaling through restoration of APC in a mouse model of CRC can reverse tumorigenesis even in the presence of mutations in other potent cancer genes such as *TP53* and *KRAS* (*Dow et al., 2015*). Hence, genes selectively required to sustain the high-level WNT signaling that ensues when *APC* or *CSNK1A1* are lost, such as those suggested by our comparative analysis, may represent potential therapeutic targets.

## SERBP1 controls CTNNB1 abundance in cells lacking APC

The second top hit of the APC suppressor screen, after *CTNNB1*, was the gene encoding the mRNA binding protein SERBP1 (*Figure 5B* and *Supplementary file 1*), also known as PAI-RBP1. *SERBP1* was also a significant hit in the CSNK1A1 suppressor screen (*Figure 5C* and *Supplementary file 1*). SERBP1 was initially identified as an mRNA binding protein that interacts with the cyclic nucleotide-responsive sequence of the Type-1 plasminogen activator inhibitor mRNA and may play a role in regulation of mRNA stability (*Heaton et al., 2001*). Yet, its cellular function remains largely unknown, and it has never been implicated in regulation of WNT signaling.

To explore the consequences of disrupting SERBP1 in cells lacking APC we used an independently isolated HAP1-7TGP clonal cell line with a lesion in the *APC* locus introduced by a GT insertion (APC$^{KO-2}$, see Materials and methods, *Supplementary file 2* and *Figure 6D*). This ensured that any effects on WNT signaling were not specific to the CRISPR/Cas9-induced lesion in the APC$^{KO-1}$

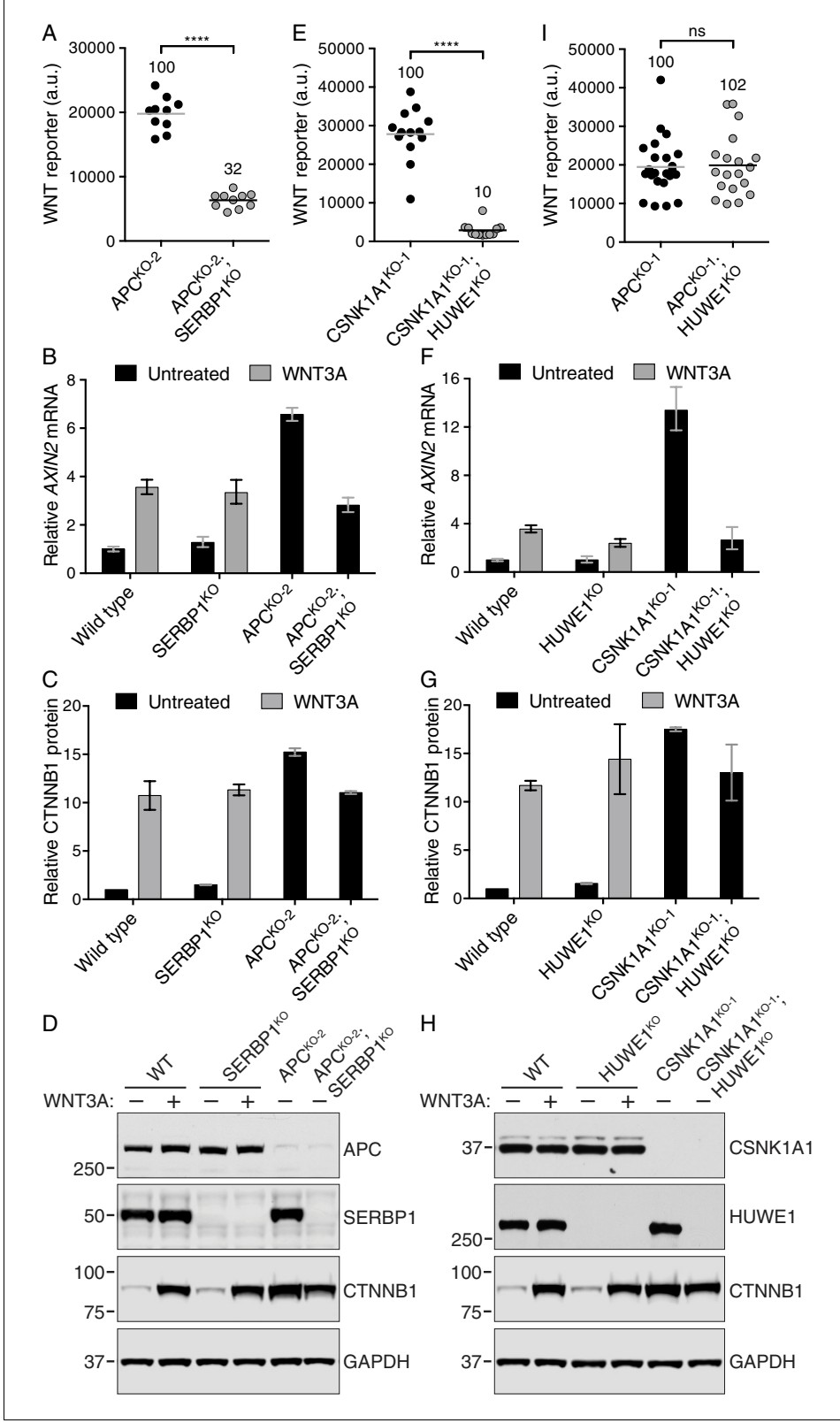

**Figure 6.** The mRNA binding protein SERBP1 controls CTNNB1 abundance in cells lacking APC (A–D), and the E3 ubiquitin ligase HUWE1 regulates WNT signaling in the absence of CSNK1A1 (E–I). (A, E, I) WNT reporter activity (median EGFP fluorescence from 5000 (A), 20,000 (E) or 2000 (I) cells) for the indicated single- and double-mutant cell lines. Each circle represents a unique clonal cell line and the average of 10 (A), ≥12 (E) or ≥19 (I) independent clones for each genotype is indicated by a horizontal line. The average percentage reporter activity relative to single-mutant cell lines is

*Figure 6 continued on next page*

*Figure 6 continued*

also indicated above each group of circles. Significance was determined by unpaired t-test with Welch's correction and is indicated as ****(*p*<0.0001) or ns (not significant). (**B,F**) *AXIN2* mRNA (average ± SD of *AXIN2* mRNA normalized to *HPRT1* mRNA, each measured in triplicate reactions), relative to untreated WT cells, for a single clonal cell line of each indicated genotype. Cells were treated with 50% WNT3A CM where indicated. The same cell lines analyzed in B were also analyzed in C and D; the same cell lines analyzed in F were also analyzed in G and H. Analysis of additional independent clonal cell lines is presented in *Figure 6—figure supplement 1*. (**C,G**) Soluble CTNNB1 protein (average ± SD of CTNNB1 intensity normalized to GAPDH intensity from duplicate immunoblots), relative to untreated WT cells. Cells were treated with 50% WNT3A CM where indicated.(**D,H**) Representative immunoblots of the indicated clonal cell lines. The CTNNB1 and corresponding GAPDH blots depicted in D and H were used for quantification in C and G, respectively. Genotypes and treatments are indicated above the blots.

The following figure supplement is available for figure 6:

**Figure supplement 1.** The mRNA binding protein SERBP1 controls CTNNB1 abundance in cells lacking APC (**A–C**), and the E3 ubiquitin ligase HUWE1 regulates WNT signaling in the absence of CSNK1A1 (**D–G**).

---

cells used for the APC suppressor screen. As expected, APC^KO-2 cells had constitutive WNT reporter expression (*Figure 5—figure supplement 1B*). We used CRISPR/Cas9 to generate multiple independent clonal cell lines derived from APC^KO-2 cells that harbored additional inactivating mutations in *SERBP1*, hereafter called APC^KO-2; SERBP1^KO cells (*Supplementary file 2* and *Figure 6D*). Disrupting *SERBP1* in cells lacking APC caused a substantial reduction in constitutive WNT reporter fluorescence, endogenous *AXIN2* mRNA and soluble CTNNB1 protein abundance (*Figure 6A–D*, and *Figure 6—figure supplement 1A and B*). Disrupting *SERBP1* in WT HAP1-7TGP cells (SERBP1^KO, *Supplementary file 2* and *Figure 6D*) did not affect basal or WNT3A-induced levels of *AXIN2* mRNA or soluble CTNNB1 (*Figure 6B–D*, and *Figure 6—figure supplement 1A and B*), explaining why *SERBP1* was not enriched for GT insertions in the WNT screen for positive regulators of ligand-induced signaling (*Figure 5D* and *Supplementary file 3*). Microinjection of *SERBP1* mRNA into *X. laevis* embryos resulted in duplication of the body axis, establishing SERBP1 as a *bona fide* positive regulator of WNT signaling in vertebrates (*Figure 2E and F*).

In the absence of destruction complex activity, SERBP1 could conceivably reduce CTNNB1 abundance by decreasing transcript or protein levels. No significant changes in *CTNNB1* mRNA levels were detected when *SERBP1* was disrupted in either WT or mutant *APC* genetic backgrounds (*Figure 6—figure supplement 1C*), suggesting instead a reduction in CTNNB1 protein. Thus, SERBP1 can regulate CTNNB1 abundance in cells lacking APC. This mechanism, independent of destruction complex activity, could be particularly well suited for therapeutic interference in tumors where APC function is lost.

## HUWE1 regulates WNT signaling in the absence of CSNK1A1

The third most significant hit of the CSNK1A1 suppressor screen, following *CTNNB1* and *CREBBP*, was *HUWE1* (*Figure 5C* and *Supplementary file 1*). *HUWE1* (also known as *MULE*, *LASU1* and *UREB1*) encodes a 480–482 kDa E3 ubiquitin ligase containing a C-terminal HECT domain with diverse cellular functions (*Bernassola et al., 2008*). Previous work has implicated HUWE1 as a negative regulator of WNT signaling (*de Groot et al., 2014*). However, the results of our screens suggested that in the absence of CSNK1A1, HUWE1 is a positive regulator of WNT signaling. To test the role of HUWE1 in this context, we further engineered the CSNK1A1^KO-1 cells used for the CSNK1A1 suppressor screen, as well as an independently derived cell line with a different lesion in *CSNK1A1* (CSNK1A1^KO-2, *Supplementary file 2*). In each parental cell line, we used two different sgRNAs to disrupt *HUWE1* and isolated multiple double-mutant clonal cell lines (designated CSNK1A1^KO-1; HUWE1^KO and CSNK1A1^KO-2; HUWE1^KO, *Supplementary file 2* and *Figure 6H*). Disruption of HUWE1 resulted in a consistent 82–90% reduction of constitutive WNT reporter fluorescence and an 80–85% reduction of endogenous *AXIN2* mRNA (*Figure 6E and F*, and *Figure 6—figure supplement 1D and E*). In addition, microinjection of *HUWE1* mRNA into *X. laevis* embryos resulted in duplication of the body axis at low frequency (*Figure 2E and F*), supporting a more general role as a positive regulator.

In contrast to the 80–90% reduction in both WNT reporter fluorescence and target gene expression (*Figure 6E and F*, and *Figure 6—figure supplement 1D and E*), depleting HUWE1 in

CSNK1A1[KO-1]; HUWE1[KO] cells reduced soluble CTNNB1 levels by only 20–32% (*Figure 6G and H*, and *Figure 6—figure supplement 1F*). These results show that in cells lacking CSNK1A1, HUWE1 has a minor influence on CTNNB1 abundance, but that its predominant role in WNT signaling is distinct from the regulation of CTNNB1 protein levels.

Additionally, *HUWE1* was not a significant hit in the WNT screens or the APC suppressor screen (*Figure 5D* and *Supplementary file 3*), suggesting that its role is specific to cells lacking CSNK1A1. Indeed, disruption of *HUWE1* in WT HAP1-7TGP cells (HUWE1[KO], *Supplementary file 2* and *Figure 6H*) did not cause significant changes in WNT3A-induced *AXIN2* mRNA or CTNNB1 protein abundance (*Figure 6F–H*, and *Figure 6—figure supplement 1E and F*). To directly test whether HUWE1 disruption reduces WNT signaling in cells lacking CSNK1A1 but not other destruction complex components, we disrupted *HUWE1* in APC[KO-1] cells (APC[KO-1]; HUWE1[KO], *Supplementary file 2*), and found no measurable defect in WNT reporter florescence (*Figure 6I*). Signaling driven by inhibition of GSK3 was also unaffected by the loss of HUWE1 in HUWE1[KO] cells (*Figure 6—figure supplement 1G*).

In summary, the drastic defect in signaling caused by loss of HUWE1 in cells lacking CSNK1A1 and its ability to promote formation of a secondary body axis when expressed ectopically in *X. laevis* embryos demonstrate a positive role for HUWE1 in WNT signaling. These effects are largely independent of changes is CTNNB1 protein abundance and are not observed when other components of the destruction complex are inactivated. From these results we conclude that CSNK1A1 regulates WNT signaling by an additional mechanism distinct from its established role in CTNNB1 turnover, and that this mechanism is mediated by HUWE1.

## Discussion

A systematic genetic analysis in human cells revealed new regulatory features at most levels of the WNT pathway, from signal reception to transcriptional activation (*Figure 7*). Based on a comparative analysis of seven genome-wide screens, we confirmed known epistatic connections and assigned new ones (*Figure 7A*). Even for some of the known WNT components, our analysis suggested unexpected regulatory mechanisms.

First, as predicted by their enrichment in the WNT screens for positive regulators (*Figure 7A*), atypical GT insertions in *AXIN2* caused an unexpected decrease in WNT signaling (*Figure 3*). These results are explained by the observation that in cells lacking the DAX domain of AXIN2, CTNNB1 is appropriately stabilized and localized to the nucleus following WNT stimulation, but remains inactive (*Figure 3*). Second, genes encoding components of the GPI anchor biosynthetic machinery, such as PIGL, and the glypican GPC4 were predominantly enriched for GT insertions in the low WNT + RSPO screen (*Figure 7A*), and we demonstrated that they indeed play a critical role in mediating signaling especially under the low WNT conditions in which RSPOs exert their strongest effect (*Figure 4*). Third, the enrichment of GT insertions in *HUWE1* only in the CSNK1A1 suppressor screen (*Figure 7A*) revealed a unique signaling condition created by disruption of CSNK1A1, but not other destruction complex components such as APC or GSK3. Only in this very specific context WNT signaling was dependent on HUWE1 (*Figure 6*). This positive regulatory function of HUWE1 is evidently different from the negative feedback regulation described previously (*de Groot et al., 2014*).

The presence of mutations in known regulators in the expected screens demonstrates the predictive power of our approach, which enabled us to infer the site of action of newly identified pathway components. The transcription factor TFAP4 would be predicted to act downstream of the CTNNB1 destruction complex based on its disruption in all screens for positive regulators of signaling (*Figure 7A*), as our experimental results confirmed (*Figure 2*). The selective disruption of *SERBP1* in only the APC and CSNK1A1 suppressor screens (*Figure 7A*) suggests a regulatory role on signaling independent of destruction complex activity, which we demonstrated experimentally (*Figure 6*).

An important conclusion from our studies is that WNT signaling can be regulated by processes other than control of CTNNB1 protein abundance by the destruction complex. We demonstrate two distinct instances in which CTNNB1 transcriptional activity can be dissociated from protein levels, one caused by truncation of the AXIN2 DAX domain and the other caused by depletion of HUWE1 in cells lacking CSNK1A1. It will be interesting to explore if these phenomena can be exploited for therapeutic purposes in tumors driven by inappropriate stabilization of CTNNB1. We also provide evidence that the destruction complex does not have a unitary function in controlling CTNNB1

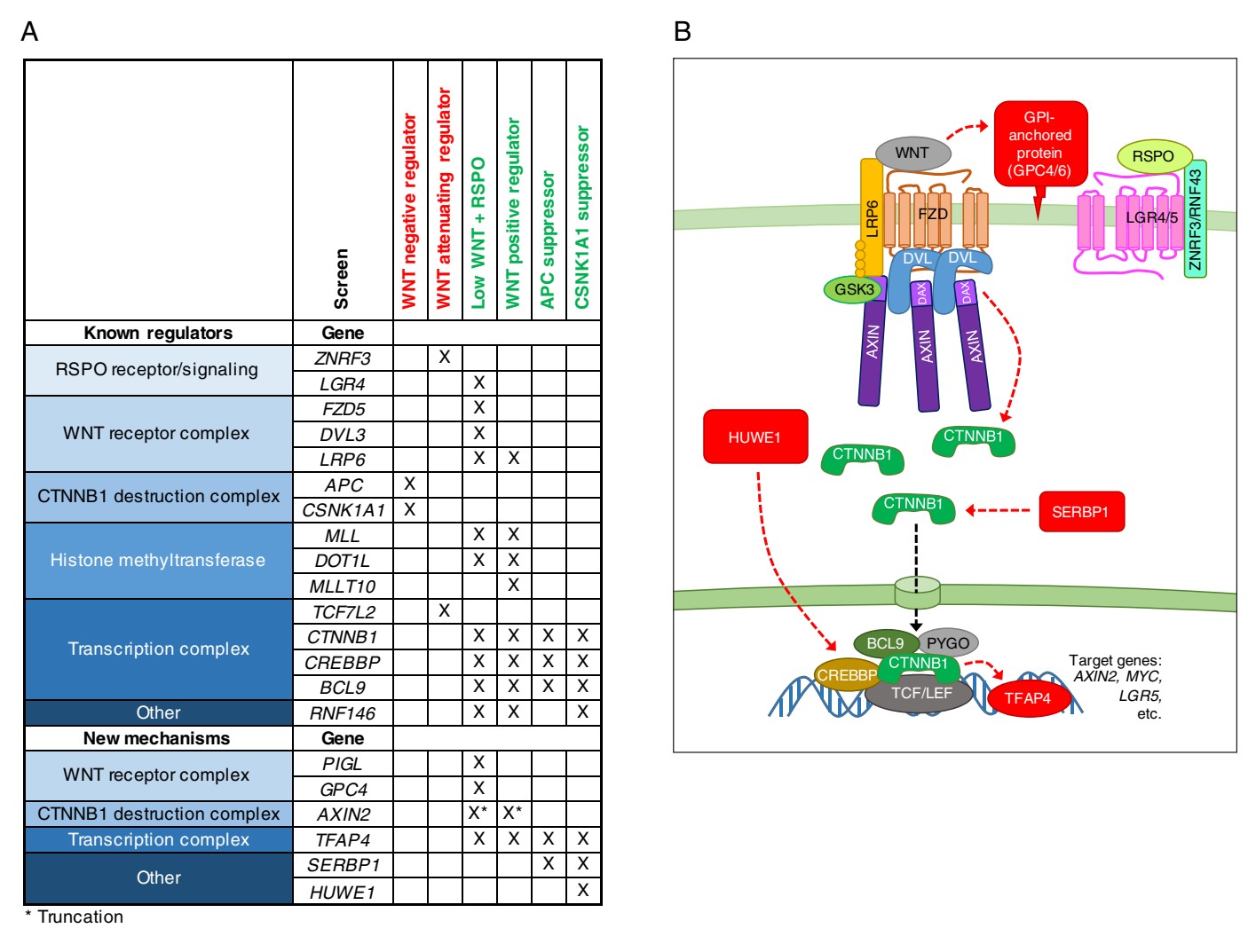

**Figure 7.** A comparative analysis of seven genome-wide screens revealed epistatic connections and regulatory mechanisms in WNT signaling. (**A**) Summary of known regulators, and new regulators or regulators mediating new mechanisms in WNT signaling validated in this study. An 'X' denotes that the gene was enriched for GT insertions in the sorted cells from the indicated genetic screen (FDR-corrected $p$-value<0.05). Known regulators are grouped into functional modules and arranged according to previously described epistatic relationships. Epistatic relationships for new regulators or regulators mediating new mechanisms are inferred based on their patterns across screens. The screens in which cells were sorted for increased WNT reporter fluorescence are labeled in red, and those in which cells were sorted for reduced WNT reporter fluorescence are labeled in green. For the 'WNT positive regulator' column, hits from the WNT screens done at both low and high stringency (*Figure 1E and F*) were considered together. For *AXIN2*, the asterisk indicates that GT insertions mapped in the sorted cells generate a dominant allele that encodes a truncated protein product. (**B**) Model of WNT/CTNNB1 signaling, highlighting in red new regulatory mechanisms uncovered and validated in this study. Red arrows represent genetic (rather than biochemical) interactions. The various proposed mechanisms are discussed throughout the Results and Discussion sections.

protein abundance, since disrupting distinct components produces different outcomes. Supressor screens in cells lacking APC or CSNK1A1 revealed mutations in substantially different sets of genes (*Figure 5*), and while SERBP1 controls CTNNB1 abundance in cells lacking APC, the effects of HUWE1 in cells lacking CSNK1A1 are largely independent of changes in CTNNB1 levels (*Figure 6*).

From these studies a more elaborate picture of the core WNT signaling cascade emerges, with additional regulation superimposed on the core module (*Figure 7B*). Further studies will be required to elucidate the mechanisms that mediate each of these new layers of regulation and to identify the

physiological or pathological contexts in which they act. Yet, the comparative analysis of seven unbiased genome-wide screens and the characterization of hits through a quantitative assessment of CRISPR/Cas9-engineered clonal cell lines provided many insights into this complex developmental signaling pathway. The conceptual and methodological framework described in this work should enable the comprehensive understanding of other signaling systems.

## Materials and methods

### Reagent providers

Reagents were obtained from the following companies: Thermo Fisher Scientific, Waltham, MA; Sigma-Aldrich, St. Louis, MO; Bio-Rad, Hercules, CA; Cell Biolabs, San Diego, CA; Clontech, Mountain View, CA; Promega, Madison, WI; GE Healthcare Life Sciences, Logan, UT; GE Dharmacon, Lafayette, CO; Addgene, Cambridge, MA; BD Biosciences, San Jose, CA; Abcam, Cambridge, MA; EMD Millipore, Billerica, MA; Bethyl Laboratories, Montgomery, TX; Santa Cruz Biotechnology, Dallas, TX; R and D Systems, Minneapolis, MN; Cell Signaling Technology, Danvers, MA; Li-Cor, Lincoln, NE; Jackson ImmunoResearch Laboratories, West Grove, PA; Developmental Studies Hybridoma Bank at the University of Iowa (DSHB), Iowa City, IA; American Type Culture Collection (ATCC), Manassas, VA; Atlanta Biologicals, Flowery Branch, GA; Pall Corporation, Fribourg, Switzerland; Selleckchem, Houston, TX; Roche, Mannheim, Germany; QIAGEN Sciences, Hilden, Germany; New England Biolabs (NEB), Ipswich, MA; Epicentre, Madison, WI; Corning, Corning, NY; Bloomington Drosophila Stock Center at Indiana University (BDSC), Bloomington, IN.

### Plasmids

7TGP was a gift from Roel Nusse (Addgene plasmid # 24305). pX330-U6-Chimeric_BB-CBh-hSpCas9 (pX330) was a gift from Feng Zhang (Addgene plasmid # 42230). pCSDest was a gift from Nathan Lawson (Addgene plasmid # 22423). pCS2$^+$, pCS2-YFP, pCS2-$\beta$-cat-S33Y, pCS2-xWnt8, VSV-G and Δ8.9 were a gift from Henry Ho.

The following plasmids were purchased: pAdVAntage (Promega Cat. # E1711), pCMV-VSV-G (Cell Biolabs Cat. # RV-110), pCMV-Gag-Pol (Cell Biolabs Cat. # RV-111), pENTR-D-TOPO (Thermo Fisher Scientific Cat. # K240020), pENTR2B (Thermo Fisher Scientific Cat. # A10463).

To generate pCSDest-TFAP4, human *TFAP4* was amplified by PCR from MGC Human Sequence-verified cDNA (Clone ID 4181538, GE Dharmacon Cat. # MHS6278-202757542) using primers hTFAP4-FOR (5'-CACCATGGAGTATTTCATGGTGCCCA-3') and hTFAP4-REV (5'- TCAGGGAAGC TCCCCGT-3'), designed to add a directional TOPO cloning sequence at the 5' end. The resulting PCR product was cloned directionally into pENTR-D-TOPO. Individual clones were screened by digestion with NotI and PstI for the presence of the insert in the correct orientation, and one clone was sequenced completely and subcloned into pCSDest using the Gateway LR Clonase II Enzyme mix (Thermo Fisher Scientific Cat. # 11791100).

To generate pCSDest-1D4-SERBP1, human *SERBP1* was amplified by PCR from MGC Human Sequence-verified cDNA (Clone ID 4477452, GE Dharmacon Cat. # MHS6278-202758125) using primers FNNterminal (5'-TTTTGAATTCGCCACCATGACCGAGACCAGCCAGGTGGCCCC TGCAGGCGGCCGGCCACCTGGGCACTTACAGGAAGG-3'), designed to add an N-terminal Kozak sequence, a 1D4 tag and a single glycine linker flanked by EcoR1 and FseI restriction sites, and RNNterminal (5'-TTTTCTCGAGGGCGCGCCTTAAGCCAGAGCTGGGAATG-3'), designed to add tandem AscI and XhoI restriction sites after the stop codon. The product was digested with EcoRI and XhoI, and subcloned into pENTR2B digested at the same sites. One clone was sequenced completely and subcloned into pCSDest using the Gateway LR Clonase II Enzyme mix.

To generate pCSDest-HUWE1, codon optimized *HUWE1* in pDONR221, a gift from Sarah J. Luchansky (Developmental and Molecular Pathways, Novartis Institutes for Biomedical Research, Cambridge, MA), was subcloned into pCSDest using the Gateway LR Clonase II Enzyme mix.

pCherry-C1 was created by replacing DsRed with mCherry RFP (*Shaner et al., 2004*) in pDsRed-monomer-C1 (Clontech Cat. # 632466). An mCherry RFP PCR fragment was obtained using primers 5'-GATCGCTAGCACCATGGTGAGCAAGGGCGAGGAGGATAAC-3' and 5'-GATCCTCGAGATCTC TTGTACAGCTCGTCCATGCCGCC-3'. The PCR product was digested with NheI and XhoI and ligated into pDsRed-monomer-C1 digested with the same enzymes.

pGT-mCherry and pGT+1mCherry retroviral gene trap (GT) vectors (diagrammed schematically in *Figure 1B*), containing an inactivated 3'LTR, a strong adenoviral (Ad40) splice acceptor site, mCherry RFP in two different reading frames following the splice acceptor site, and the SV40 polyadenylation signal, were created by replacing GFP with mCherry RFP in pGT-GFP (*Carette et al., 2009*). To create pGT-mCherry, the following primers were used to generate a PCR product containing the splice acceptor site followed by mCherry: 5'-GATCATCGATGCGCAGGCGCAATCTTCGCATTTC TTTTTTCCAGATGGTGAGCAAGGGCGAGG-3' and 5'-GATCGGATCCTTACTTGTACAGCTCGTCCA TG-3'. To create pGT+1mCherry, the following primers were used: 5'-GATCATCGATGCGCAGGCG-CAATCTTCGCATTTCTTTTTTCCAGGATGGTGAGCAAGGGCGAGG-3' and 5'-GATCGGATCCTTAC TTGTACAGCTCGTCCATG-3'. The PCR products were digested with ClaI and BamHI and cloned into pGT-GFP digested with the same enzymes.

All constructs were confirmed by sequencing.

## Antibodies

### For immunoblotting

Primary antibodies: mouse anti-$\beta$-catenin (Clone 14/Beta-Catenin) (1:1000, BD Biosciences Cat. # 610154), mouse anti-GAPDH [6 C5] (1:5000, Abcam Cat. # ab8245), mouse anti-ACTIN (clone C4) (1:500, EMD Millipore Cat. # MAB1501), rabbit anti-AP4 (TFAP4) serum (1:2000, a gift from Takeshi Egawa; *Egawa and Littman, 2011*), rabbit anti-SERBP1 (1:5000, Bethyl Laboratories Cat. # A303-938A), goat anti-casein kinase I$\alpha$ (C-19) (1:200, Santa Cruz Biotechnology Cat. # sc-6477), rabbit anti-APC (C-20) (1:500, Santa Cruz Biotechnology Cat. # sc-896), rabbit anti-Lasu1/Ureb1 (HUWE1) (Bethyl Laboratories Cat. # A300-486A), goat anti-AXIN1 (1:1250, R and D Systems Cat. # AF3287), rabbit anti-AXIN2 (76 G6) (1:500, Cell Signaling Technology Cat. # 2151), mouse anti-V5 (1:5000, Thermo Fisher Scientific Cat. # R960CUS (formerly 46–1157)), mouse anti-$\alpha$-Tubulin (clone DM1A) (1:10,000, Sigma-Aldrich Cat. # T6199).

Secondary antibodies: IRDye 800CW donkey anti-mouse IgG (H+L) (1:10,000, Li-Cor Cat. # 926–32212), IRDye 680RD donkey anti-rabbit IgG (H+L) (1:10,000, Li-Cor Cat. # 925–68073), peroxidase AffiniPure donkey anti-goat IgG (H+L) (1:5000, Jackson ImmunoResearch Laboratories Cat. # 705-035-003), peroxidase AffiniPure goat anti-rabbit IgG (H+L) (1:7500, Jackson ImmunoResearch Laboratories Cat. # 111-035-003), peroxidase AffiniPure donkey anti-mouse IgG (H+L) (1:5000, Jackson ImmunoResearch Laboratories Cat. # 715-035-150), goat anti-mouse IgG (H+L) HRP conjugate (1:10,000, Bio-Rad Cat. # 1706516).

Primary and secondary antibodies used for detection with the Li-Cor Odyssey imaging system were diluted in a 1 to 1 mixture of Odyssey Blocking Buffer (Li-Cor Cat. # 927–40000) and TBST (Tris buffered saline (TBS) + 0.1% Tween-20), and those used for detection by chemiluminescence were diluted in TBST + 5% skim milk. All primary antibody incubations were done overnight at 4°C, and secondary antibody incubations were done for 1 hr at room temperature (RT).

### For immunostaining

Primary antibodies: mouse anti-$\beta$-catenin (Clone 14/Beta-Catenin) (1:500, BD Biosciences Cat. # 610154), mouse anti-Wingless (1:25, DSHB Cat. # 4D4 (supernatant) for *D. melanogaster* wing imaginal discs and 1:200, DSHB Cat. # 4D4 (concentrate) for embryos), guinea pig anti-Senseless (1:1000; *Nolo et al., 2000*), mouse anti-Engrailed/Invected (1:100, DSHB Cat. # 4D9 (concentrate)).

Secondary antibodies: donkey anti-mouse IgG (H+L) Alexa Fluor 647 conjugate (1:250, Thermo Fisher Scientific Cat. # A-31571), donkey anti-mouse IgG (H+L) Alexa Fluor 555 conjugate (1:400, Thermo Fisher Scientific Cat. # A-31570), goat anti-guinea pig IgG (H+L) highly cross-adsorbed, Alexa Fluor 488 conjugate (1:400, Thermo Fisher Scientific Cat. # A-11073).

## Cell lines and growth conditions

L Wnt-3A (ATCC Cat. # CRL-2647), L cells (ATCC Cat. # CRL-2648), 293T (ATCC Cat. # CRL-3216) and 293FT cells (Thermo Fisher Scientific Cat. # R70007) were grown at 37°C and 5% $CO_2$ in complete growth medium (CGM) 1: Dulbecco's Modified Eagles Medium (DMEM) with High Glucose, without L-Glutamine and Sodium Pyruvate (GE Healthcare Life Sicences Cat. # SH30081.01); 1X GlutaMAX-I (Thermo Fisher Scientific Cat. # 35050079); 1X MEM Non-Essential Amino Acids (Thermo Fisher Scientific Cat. # 11140050); 1 mM Sodium Pyruvate (Thermo Fisher Scientific Cat. #

11360070); 40 Units/ml Penicillin, 40 µg/ml Streptomycin (Thermo Fisher Scientific Cat. # 15140122); 10% Fetal Bovine Serum (FBS) (Atlanta Biologicals Cat. # S11150). HAP1 haploid human cells (kindly provided by Thijn Brummelkamp, now available from Horizon Discovery, Cambridge, United Kingdom) were derived and characterized as described previously (*Carette et al., 2011b*). Throughout the course of experiments, the ploidy of HAP1 cells and derivatives thereof was routinely tested by DNA content analysis of propidium iodide (PI)-stained nuclei, as described below. Genetically modified clonal derivatives were confirmed by sequencing of target loci and in some cases immunoblotting, as described below (see also *Supplementary file 2*). HAP1 cells and derivatives thereof were grown at 37°C and 5% $CO_2$ in CGM 2: Iscove's Modified Dulbecco's Medium (IMDM) with L-glutamine, with HEPES, without Alpha-Thioglycerol (GE Healthcare Life Sciences Cat. # SH30228.01); 1X GlutaMAX-I; 40 Units/ml Penicillin, 40 µg/ml Streptomycin; 10% FBS.

## Preparation of WNT3A and L cell conditioned media

L Wnt-3A cells or L cells were seeded in 15 cm tissue culture-treated dishes at a density of $1.5 \times 10^6$ cells per dish and grown in 25 ml of CGM 1. After 3 days, the medium was refreshed, and after an additional 3 days, WNT3A or L cell conditioned medium (used as a control) was collected, filtered through a 0.2 µm polyethersulfone (PES) membrane filter, aliquoted, flash-frozen in liquid nitrogen and stored at −80°C. The medium was thawed immediately before use, and leftover medium was used only after one additional freeze-thaw cycle.

## Construction of HAP1-7TGP WNT reporter haploid cell line

Lentivirus containing the 7TGP WNT reporter construct (*Fuerer and Nusse 2010*) was produced in 293T cells. ~24 hr before transfection, 293T cells were seeded in a 10 cm dish at a density of $4 \times 10^6$ per dish and grown in 10 ml of CGM 1. When the cells were nearly confluent, the medium was replaced with CGM 1 without antibiotics. To prepare a calcium phosphate transfection mix, 8 µg of 7TGP plasmid, 4 µg of VSV-G, 4 µg of Δ8.9 and 0.5 µg of pCS2-YFP (used as a co-transfection marker) in 450 µl of sterile water were mixed with 50 µl of 2.5 M $CaCl_2$. The DNA/$CaCl_2$ solution was added to 500 µl of 2X HBS (42 mM HEPES pH 7.04, 274 mM NaCl, 10 mM KCl, 15 mM dextrose, 1.4 mM $Na_2HPO_4.7H_2O$) and mixed by bubbling air through the solution. Following a 20 min incubation at RT, the transfection mix was added drop-wise to the dish of cells and mixed by gentle agitation. The medium was refreshed 16 hr post-transfection, at which time the efficiency of transfection was assessed by microscopic inspection of YFP fluorescence. Lentivirus-containing medium was collected 36, 48 and 60 hr post-transfection (30 ml total), filtered through 0.45 µm filters (Acrodisc syringe filters with Supor membrane, Pall Corporation) and concentrated by ultracentrifugation for 1.5 hr at 23,000 rpm in a Sorvall Surespin 630 rotor. The supernatants were discarded, and the pellets overlaid by a total of 200 µl of sterile phosphate buffered saline (PBS) supplemented with 1 mg/ml bovine serum albumin (BSA). Following 12 hr of incubation at 4°C, the pellets were resuspended, aliquoted, flash-frozen in liquid nitrogen, and stored at −80°.

Approximately 48 hr before transduction, HAP1 cells were seeded in 10 cm dishes at a density of $1.5 \times 10^6$ per dish and grown in 10 ml of CGM 2. Cells were transduced with 145 µl of freshly thawed lentivirus diluted in 10 ml of CGM 2 supplemented with 4 µg/ml polybrene. ~48 hr post-transduction cells were treated with 1 µg/ml puromycin to select for those with a stably integrated 7TGP cassette. After selection was complete (assessed by death of >99% of control, untransduced HAP1 cells), surviving cells were treated for 16–24 hr with 50% WNT3A CM in CGM 2, and single WNT3A-responsive cells exhibiting the highest ~50% EGFP fluorescence were sorted by FACS into 96-well plates containing 200 µl of CGM 2 per well. After 14 days of undisturbed growth, individual clones were amplified and haploid HAP1-7TGP clonal cell lines were identified by DNA content analysis of PI-stained nuclei (*Nicoletti et al., 1991*): $1.5 \times 10^6$ cells were incubated with 300 µl of hypotonic fluorochrome solution (50 µg/ml PI, 0.1% sodium citrate, 0.1% Triton X-100) for >15 min at RT and PI fluorescence was measured by FACS in a BD LSRFortessa cell analyzer (BD Biosciences) using a 561 laser and 600 LP, 610/20 BP filters. A haploid cell line with low basal reporter activity and a high dynamic range of EGFP fluorescence in response to WNT3A (*Figure 1—figure supplement 1A*) was expanded and used in all subsequent studies.

## Analysis of WNT reporter fluorescence

To measure WNT reporter activity in HAP1-7TGP cells or derivatives thereof, ~24 hr before treatment cells were seeded in 24-well plates at a density of $8 \times 10^4$ per well and grown in 0.5 ml of CGM 2. Cells were treated for 16–24 hr with the indicated concentrations of WNT3A CM, L cell CM, recombinant mouse WNT3A (R and D Systems Cat. # 1324-WN) recombinant human RSPO1 (R and D Systems Cat. # 4645-RS), LiCl, CHIR-99021 (CT99021) (Selleckchem Cat. # S2924) or XAV-939 (Selleckchem Cat. # S1180) diluted in CGM 2. Cells were washed with 0.5 ml PBS, harvested in 150 µl of 0.05% Trypsin-EDTA (0.05%) (Thermo Fisher Scientific Cat. # 25300054), resuspended in 450 µl of CGM 2, and EGFP fluorescence was measured immediately by FACS on a BD LSRFortessa cell analyzer (BD Biosciences) using a 488 laser and 505 LP, 530/30 BP filters, or on a BD Accuri RUO Special Order System (BD Biosciences). Typically, fluorescence data for 5000–20,000 singlet-gated cells was collected and, unless indicated otherwise, the median EGFP fluorescence ± standard error of the median (SEM = $1.253 \, \sigma/\sqrt{n}$, where $\sigma$ = standard deviation and $n$ = sample size) was used to represent the data.

## Reporter-based forward genetic screens

### Mutagenesis of haploid cells with GT retrovirus

HAP1-7TGP cells or derivatives thereof were mutagenized as described previously (*Carette et al., 2011a*), with some modifications. GT-containing retrovirus was produced in 293FT cells. ~24 hr before transfection, a total of $9 \times 10^7$ cells were seeded in 6 T-175 flasks ($1.5 \times 10^7$ per flask), each containing 30 ml of CGM 1 without antibiotics. The medium in each T-175 flask was refreshed immediately before transfection, and cells were transfected with a transfection mix prepared as follows: 3.3 µg each of pGT-mCherry and pGT+1mCherry, 1.7 µg pAdVAntage, 2.6 µg pCMV-VSV-G and 4 µg pCMV-Gag-Pol were prepared in 450 µl serum-free DMEM (GE Healthcare Life Sicences Cat. # SH30081.01), and 45 µl of the X-tremeGENE HP DNA transfection reagent (Roche Cat. # 06366546001) were added and incubated for 20 min at RT. The medium was refreshed ~24 hr post-transfection, and retrovirus-containing supernatant was collected ~48 and ~56 hr post-transfection. The virus-containing supernatant was immediately filtered through 0.45 µm filters and concentrated by ultracentrifugation for 1.5 hr at 23,000 rpm in a Beckman SW 32 Ti or a Sorvall Surespin 630 rotor. Most of the supernatant was removed, leaving behind a small volume to cover the pellet. The virus was incubated at 4°C for ~12 hr, resuspended in CGM 2, pooled and supplemented with 4 µg/ml polybrene.

Approximately $1.2 \times 10^8$ HAP1-7TGP, APC$^{KO-1}$ or CSNK1A1$^{KO-1}$ cells were mutagenized by transduction with retroviral supernatant. ~24 hr before transduction, $6 \times 10^7$ cells sere seeded in 3 T-175 flasks ($2 \times 10^7$ per flask), each containing 30 ml of CGM 2. All the freshly prepared retrovirus was divided into three equal parts, and added to each of the three flasks of cells in two doses ~8 hr apart. Cells were amplified by passaging every ~48 hr, and during each passage cells from all flasks were pooled and $1.2 \times 10^8$ were re-seeded in order to retain the complexity of the mutagenized library. $3 \times 10^7$ mutagenized cells were used to generate a control dataset of GT insertions (see below).

### FACS-based phenotypic enrichment

Screens were started at least 6 days following mutagenesis; for some screens, mutagenized cells were frozen 3 days following transduction with GT-containing retrovirus, and allowed to recover for at least 3 days after thawing and before starting treatments for the screens. ~24 hr before treatment, a total of $1.25 \times 10^8$ cells were seeded in 5 T-175 flasks ($2.5 \times 10^7$ per flask), and grown in 30 ml of CGM 2. Cells were treated for 16–24 hr as follows: for the WNT negative regulator screen (*Figure 1C*), and for the APC (*Figure 5B*) and CSNK1A1 (*Figure 5C*) suppressor screens, cells were left untreated; for the WNT attenuating regulator screen (*Figure 1D*) cells were treated with 12.5% WNT3A CM; for the WNT positive regulators screens at low (*Figure 1E*) and high (*Figure 1F*) stringencies, cells were treated with 50% WNT3A CM; for the low WNT + RSPO screen (*Figure 4A*), cells were treated with 1.04% WNT3A CM and 10 ng/ml recombinant human RSPO1. Cells were harvested, concentrated to ~$2 \times 10^7$ per ml, strained through a 70 µm cell strainer and a minimum of $2.5 \times 10^7$ singlet-gated cells from each flask were sorted in a BD FACSAria Special Order Research Product (BD Biosciences) cell sorter through a 70 µm nozzle, using gates encompassing the

indicated levels of WNT reporter (EGFP) fluorescence. In this way, a total of $1.25 \times 10^8$ cells from five flasks were sorted, retaining the diversity of the initial mutant library. A population of singlet-gated cells (not gated based on WNT reporter fluorescence) was also sorted and carried along as a control to set FACS gates during subsequent sorts.

Cells were expanded for 6–8 days to allow recovery and resetting of the WNT reporter, and a portion of the cells (three to four times the number of cells enriched for during the prior sort, in order to maintain an over-representation of the library diversity) was used in the subsequent round of sorting, following the same treatments and FACS gating criteria described above for each screen. For the WNT negative (*Figure 1C*), attenuating (*Figure 1D*), and positive regulator screens at low (*Figure 1E*) and high (*Figure 1F*) stringencies, as well as for the APC (*Figure 5B*) and CSNK1A1 (*Figure 5C*) suppressor screens, cells underwent a total of two consecutive rounds of FACS sorting. For the low WNT + RSPO screen (*Figure 4A*), cells underwent a total of four consecutive rounds of FACS sorting. $3 \times 10^7$ cells expanded following the final sort for each screen were used to generate a dataset of GT insertions in the sorted cell populations (see below).

## Preparation of genomic GT insertion libraries for deep sequencing

To map GT insertions in control and sorted cells, sequencing libraries were prepared using a linear amplification (LAM) PCR-mediated method described previously (*Carette et al., 2011a*), with modifications. Steps prior to and following each of the two PCR amplifications were carried out in separate dedicated lab spaces to avoid cross-contamination. Genomic DNA was extracted using the QIAamp DNA Mini Kit (QIAGEN Cat. # 51304) according to the manufacturer's 'Spin Protocol: DNA Purification from Blood or Body Fluids'. Each $3 \times 10^7$ cell pellet was resuspended in 600 μl PBS, split into three columns, and eluted in 100 μl ultrapure water per column. Typically, 100–130 μg of total genomic DNA were recovered.

A total of 40 μg of DNA were used for the subsequent step, in which 20 μg of DNA were separately digested with 100 units of MseI (NEB Cat. # R0525L) or SpeI (NEB Cat. # R0133L) in a 100 μl reaction containing 1X CutSmart buffer, with overnight incubation at 37°C. DNA from each digest was purified with the QIAquick PCR Purification Kit (QIAGEN Cat. # 28104) according to the manufacturer's protocol, using two separate columns and eluting from each column in 50 μl ultrapure water. DNA recovered from both digests was pooled to obtain approximately 20–30 μg.

A total of 14 μg of DNA were used as template in the subsequent LAM-PCR reaction, initiated from a 5′ biotinylated primer specific to a unique viral packaging signal in the GT sequence, and elongating through the adjacent proviral LTR into the flanking sequence in the genome. Seven replicate reactions were set up, each containing 2 μg of DNA, 0.5 pmol of the primer 3LTRflanking-biot1 (5′-GGTCTCCAAATCTCGGTGGAAC-3′), 1X AccuPrime PCR Buffer II and 0.4 μl AccuPrime *Taq* DNA Polymerase, High Fidelity (Thermo Fisher Scientific Cat. # 12346) in a total volume of 50 μl. The following PCR program was run: (5 min at 95°C) x 1 cycle; (30 s at 95°C, 30 s at 58°C, 60 s at 68°C) x 100 cycles. After the first 51 cycles, the reaction was paused and another 0.4 μl of AccuPrime *Taq* DNA Polymerase were added to each reaction. A control reaction containing 2 μg of digested DNA from unmutagenized cells lacking the provirus was run in parallel.

The resulting PCR product from each replicate LAM-PCR reaction was purified on 8 μl of magnetic Dynabeads M280 (Dynabeads kilobaseBINDER Kit, Thermo Fisher Scientific Cat. # 60101). The beads were prewashed three times with 100 μl PBS and once with 20 μl of the Binding Solution included in the kit, using a multi-well magnet. The beads were resuspended in 50 μl Binding Solution, and each 50 μl LAM-PCR reaction was added and mixed thoroughly to resuspend the beads. Reactions were incubated overnight at RT with gentle rotation to capture the biotinylated PCR product on the streptavidin-coated beads.

A 5′ phospho, 3′ dideoxy DNA linker (5′-TCGTATGCCGTCTTCTGCTTGACTCAGTAGTTG TGCGATGGATTGATG-3′) complementary to a Solexa adaptor sequence (see below) was appended to the 3′ end of each amplicon using the CircLigase II ssDNA ligase (Epicentre, Cat. # CL9025K). The 3′ dideoxy modification on the linker prevents self-ligation, while the biotinylated 5′ end of the LAM-PCR product prevents amplicon multimerization. The beads were washed three times with 100 μl PBS (excess buffer was carefully removed), resuspended thoroughly in 20 μl of a reaction containing 25 pmol of DNA linker, 1X CircLigase II reaction buffer, 2.5 mM MnCl$_2$, 1 M Betaine and 0.75 μl CircLigase II ssDNA Ligase, and incubated at 60°C for 1 hr.

The LAM-PCR product comprising genomic sequence flanked by the LTR junction on the 5' end and by the linker on the 3' end was amplified exponentially using complementary primers containing Solexa adaptor sequences, LTRSolexaI (5'-AATGATACGGCGACCACCGAGATCTGATGGTTCTC TAGCTTGCC-3') and Solexa adaptor II (5'-CAAGCAGAAGACGGCATACGA-3'). The beads were washed three times with 100 µl PBS (excess buffer was carefully removed) and resuspended thoroughly in 50 µl of a reaction containing 25 pmol of each primer, 1X AccuPrime PCR Buffer II and 0.6 µl of AccuPrime *Taq* DNA Polymerase, High Fidelity. The following PCR program was run: (2 min at 95°C) x 1 cycle; (30 s at 95°C, 30 s at 55°C, 90 s at 68°C) x 35 cycles; (7 min at 68°C) x 1 cycle. Tubes were centrifuged briefly to recover the solution and pellet the beads, and the supernatant was removed for analysis and final PCR product purification.

8 µl of the PCR product was analyzed on a 2% agarose diagnostic gel, along with the control reaction prepared from unmutagenized cells. A smear between 100 and 500 base pairs (bp) is typically visible for reactions from GT-mutagenized DNA, indicative of a successful library preparation, but not for the control reaction from unmutagenized DNA. Successful reactions were pooled, and the final PCR product from 100 µl of solution was purified using the QIAquick PCR Purification Kit and eluted in 50 µl of ultrapure water. A 10 nM solution of the product (the concentration was calculated using an estimated average PCR product size of 360 bp) was sequenced on one lane of an Illumina Genome Analyzer IIx using a custom sequencing primer specific to the outer extreme end of the LTR (SolexaseqFlank: 5'-CTAGCTTGCCAAACCTACAGGTGGGGTCTTTCA-3'), resulting in 36 bp reads of the sequence directly flanking GT insertion sites. FASTQ files containing the sequencing data for unsorted (control) and sorted cells from each screen have been deposited in the National Center for Biotechnology Information (NCBI) Sequence Read Archive (SRA) with Study accession number SRP094861.

## Mapping of sequencing reads and statistical analysis of GT insertion enrichment

GT insertions were mapped as described previously (*Carette et al., 2011a*). Briefly, the 36 bp sequences in the FASTQ sequencing data files were aligned to the human genome version 18 (hg18) using Bowtie alignment software (*Langmead et al., 2009*). To generate a dataset of unique GT insertion sites, we excluded ambiguous alignments (no mismatches in the full 36 bp sequence were allowed) and sequences that aligned non-uniquely to the human genome. Furthermore, the sequence was required to align uniquely to the human genome even when 1 or 2 mismatches were allowed, and if two sequences aligned 1 or 2 base pairs apart, only one was retained.

For each screen, insertions mapped to genes in the sorted cells were compared to those in the unsorted control population. The number of insertions per gene was determined by intersecting each dataset of unique insertion sites with a list of the chromosomal coordinates of RefSeq-annotated genes. If an insertion was located in a genomic region shared by multiple transcripts of the same gene, it was only counted once. GT insertions in annotated genes were classified as being in the sense or antisense orientation relative to the coding sequence of the gene, and as being in exons or introns. For the unsorted (control) cells, the number of GT insertions in each gene (regardless of orientation), and the total number of insertions for all genes was determined. For the sorted cells, the number of inactivating GT insertions in each gene, defined as sense and antisense insertions in exons and sense insertions in introns, and the total number of inactivating insertions for all genes was determined. Enrichment of GT insertions for a given gene was calculated by comparing the relative frequency at which that gene harbored *inactivating* GT insertions in the sorted cells compared to the relative frequency at which the gene carried *any* GT insertion in the control dataset. For each gene, a *p*-value was calculated using the one-sided Fisher exact test and was corrected for false discovery rate (FDR) (see *Supplementary file 1*).

## Comparative analysis of significant hits across screens

The FDR-corrected *p*-value for enrichment of GT insertions in the sorted compared to the control cells depends on many experimental variables affecting both datasets, making it an equivocal metric to directly compare hits between screens. We therefore developed an additional scoring metric that depends only on the dataset for the sorted cells. It relies on the fact that generally only sense GT insertions in introns should inactivate genes due to the directionality of the splice acceptor in the

GT, and it captures the relative abundance of intronic GT insertions in the sense and antisense orientations, as well as the overall number of intronic insertions. We defined the Intronic GT Insertion Orientation Bias (IGTIOB) score as $\log_2(S/A) \times \ln(S \times A)$, where 'S' and 'A' equal one plus the number of unique sense or antisense intronic GT insertions, respectively, in a given gene. High, positive IGTIOB scores generally indicate genes whose disruption promotes the phenotype enriched for during the screen.

To compare hits between screens, we first generated a list of genes including only those with a stringent GT enrichment FDR-corrected *p*-value<$10^{-4}$ in at least one of the screens being considered *Supplementary file 3.* We then used the IGTIOB scores of those genes in all the relevant screens to build a heat map (*Figures 4B* and *5D*). Genes were clustered using the absolute value/city block setting and complete linkage method (without normalization) in the hierarchical clustering tool of the Partek Genomics Suite software, and the data range was adjusted to encompass only non-negative IGTIOB scores (i.e. negative values were displayed as 0). Groups of genes preferentially enriched for GT insertions in one or more screens (as determined by visual inspection) are indicated next to the heat maps.

## Construction of mutant HAP1-7TGP cell lines by CRISPR/Cas9-mediated genome editing

Oligonucleotides encoding single guide RNAs (sgRNAs) (*Supplementary file 4*) were selected from a published library (*Shalem et al., 2014*), or designed using either of two online CRISPR design tools (*Hsu et al., 2013*; *Doench et al., 2014*) and cloned into pX330 according to a published protocol (original version of 'Target Sequence Cloning Protocol' from http://www.genome-engineering.org/crispr/wp-content/uploads/2014/05/CRISPR-Reagent-Description-Rev20140509.pdf; *Cong et al., 2013*).

Clonal HAP1-7TGP cell lines were established by transient transfection with pX330 containing the sgRNA followed by single cell sorting as follows. A transfection mix was prepared by diluting 450 ng of pX330 and 50 ng of pmCherry (used as a co-transfection marker for FACS sorting) in 48 µl Opti-MEM I, adding 2 µl of X-tremeGENE HP and incubating for 20 min at RT. HAP1-7TGP cells or derivatives thereof were reverse-transfected in a well of a 24-well plate by overlaying 0.5 ml of CGM 2 (without antibiotics) containing $6 \times 10^5$ cells over the 50 µl of transfection mix. Cells were passaged to a 10 cm dish ~24 hr post-transfection, using 150 µl of Trypsin-EDTA (0.25%) (Thermo Fisher Scientific Cat. # 25200056) to detach them (reverse-transfection of HAP1 cells caused unusually high adherence, hence the higher trypsin concentration). Four to five days post-transfection, single transfected (mCherry-positive) cells were sorted into 96-well plates containing 200 µl of CGM 2 per well and grown undisturbed for 16 to 17 days. Single colonies were passaged to 24-well plates, and a small number of cells was reserved for genotyping.

For genotyping, genomic DNA was extracted by adding 4 volumes of QuickExtract DNA Extraction Solution (Epicentre Cat. # QE09050) to the cells. Extracts were incubated 10 min at 65°C, 3 min at 98°C, and 5 µl were used as input for PCR amplification of the sgRNA target site in 15 µl reactions containing 1X LongAmp *Taq* reaction buffer, 300 µM of each dNTP, 400 nM of each of the flanking primers indicated in *Supplementary file 4* (most of them designed using the Primer-BLAST online tool from the NCBI) and 0.1 units/µl of LongAmp *Taq* DNA polymerase (NEB Cat. # M0323L). The presence of desired mutations was initially assessed by analysis of the PCR products on a 1% agarose gel (i.e. the absence of an amplicon or a shift in its size was deemed indicative of a lesion) and was confirmed by sequencing the amplicons using the primers indicated in *Supplementary file 4*. Given that most engineered cell lines remained haploid, sequencing results were usually unequivocal. A summary of the sequencing results for all clonal cell lines used in the study is presented in *Supplementary file 2*, and for selected clonal cell lines immunoblot analysis of the protein products is presented in *Figures 2C*, *6D and H*, *Figure 3—figure supplement 1A* and *Figure 5—figure supplement 1A*.

Whenever possible, multiple independent mutant cells lines, often generated using two different sgRNAs (see *Supplementary file 2*), were expanded and used for further characterization. For some of the comparisons between WT and mutant cells, multiple individual cell lines confirmed by sequencing to be WT at the sgRNA target site were also expanded and used as controls. To generate double and triple mutant cell lines, a single clonal cell line with the first desired mutation was used in a subsequent round of transfection with pX330 containing the second and, if applicable,

third sgRNAs. Alternatively, WT HAP1-7TGP cells were directly transfected with a combination of pX330 constructs targeting two genes simultaneously. To facilitate screening of mutant clones by PCR when targeting two genes simultaneously, we sometimes targeted one of them at two different sites within the same exon or on adjacent exons and amplified genomic sequence encompassing both target sites. Mutant clones were readily identified by the reduced size of the resulting ampli-con, and the precise lesion was confirmed by sequencing.

## Isolation of APC[KO-2] mutant cell line containing a GT insertion

As an alternative to CRISPR/Cas9-mediated genome editing, a mutant cell line containing a GT insertion in APC was isolated as follows. Mutagenized HAP1-7TGP cells enriched during the WNT-negative regulator screen (*Figure 1C*) were used to isolate the APC[KO-2] clone. Following the screen, the same FACS gate used during the screen was used to sort single cells into 96-well plates.

Colonies were harvested after 16 days, 1/10th of each clone was passaged for continued growth, and the remainder of the cells from adjacent pairs of rows in each plate were pooled. A portion of all the cells from pairs of plates were further pooled so as to obtain 'plate-pair' pools, and 'row-pair' sub-pools. Cells from plate-pair pools were harvested by centrifugation, genomic DNA was prepared using the QIAamp DNA mini kit and each pool was probed for clones containing GTs in *APC* using a nested PCR strategy. A genomic region of *APC* enriched for GT insertions was amplified by PCR using a forward primer complementary to a unique sequence in the GT (pGT-Puro4: 5'-TC TCCAAATCTCGGTGGAAC-3') and a reverse primer complementary to a unique genomic sequence adjacent to the GT-enriched region in *APC* (APC_GT: 5'-TGCTACAATGAGCTGTTAAAATGG-3'). 400 ng of genomic DNA was used as input for PCR amplification in 25 µl reactions containing 1X LongAmp *Taq* reaction buffer, 300 µM of each dNTP, 400 nM of each primer and 0.1 units/µl of LongAmp *Taq* DNA polymerase. For each plate-pair pool, the presence of clones containing a GT insertion was evident as discrete bands when the PCR products were analyzed on a 1% agarose gel.

Once positive plate-pair pools were identified, genomic DNA from the corresponding row-pair sub-pools was prepared using QuickExtract DNA Extraction Solution and the PCR procedure was repeated to identify row pairs containing clones with GT insertions. Finally, the individual clonal cell lines that had been passaged were harvested from the 96-well plates containing GT-positive row pairs, and GT-containing clones were identified using the same procedure. To map the precise genomic location of the GT insertion in APC[KO-2] (see *Supplementary file 2*), the final PCR product obtained from an individual GT-positive clone was sequenced.

## Rescue of TFAP4[CR-1] cells

Rescue of TFAP4[CR-1] cells and overexpression of TFAP4 in WT HAP1-7TGP cells (*Figure 2B*) was done by transient transfection. TFAP4[CR-1] and HAP1-7TGP were reverse-transfected in a well of a 24-well plate by overlaying 0.5 ml of CGM 2 (without antibiotics) containing $6 \times 10^5$ cells over 50 µl of a transfection mix containing 450 ng of pCSDest-TFAP4 or pCS2[+] (vector control) and 50 ng of pmCherry (as a co-transfection marker for FACS analysis) in Opti-MEM I, and 2 µl of X-tremeGENE HP. Cells were harvested using 150 µl of Trypsin-EDTA (0.25%) and passaged to a 6 cm dish ~24 hr post-transfection. ~48 hr post-transfection cells were treated for 16–24 hr with 50% WNT3A CM where indicated, and the WNT reporter (EGFP) fluorescence of mCherry-positive, singlet-gated cells was measured by FACS.

## Epistasis analysis in TFAP4[CR-1] cells

Epistasis analysis in WT HAP1-7TGP and TFAP4[CR-1] cells (*Figure 2D*) was done following treatment with WNT3A, the GSK3 inhibitor LiCl, or following transient transfection with dominant negative CTNNB1 (S33Y mutant) as follows. WT HAP1-7TGP and TFAP4[CR-1] cells were treated for 16–24 hr with 50% WNT3A CM or with 40 mM LiCl in CGM 2 where indicated, and the WNT reporter fluorescence of singlet-gated cells was measured by FACS. Alternatively, WT HAP1-7TGP and TFAP4[CR-1] cells were reverse transfected with pCS2[+] (vector control) or pCS2-$\beta$-cat-S33Y and pmCherry (as a co-transfection marker for FACS analysis) as described in the previous section. Cells were passaged ~24 hr post-transfection, and ~48 hr post-transfection the WNT reporter (EGFP) fluorescence of mCherry-positive, singlet-gated cells was measured by FACS.

## Quantitative RT-PCR analysis

Approximately 24 hr before treatment, cells were seeded in 24-well plates at a density of $2 \times 10^5$ per well and grown in 0.5 ml of CGM 2. Cells were treated for 12 hr with 50% WNT3A CM in CGM 2 where indicated. The medium was removed, and cells were harvested without washing in 800 µl of TRIzol Reagent (Thermo Fisher Scientific Cat. # 15596018). Extracts were processed according to the manufacturer's protocol taking the appropriate precautions to avoid contamination with nucleases, and total RNA was resuspended in 40 µl of DEPC-treated water (Thermo Fisher Scientific Cat. # AM9920). To synthesize cDNA, 250 ng of RNA were diluted in 8 µl DEPC-treated water, 2 µl of 5X iScript Reverse Transcrition Supermix for RT-qPCR (Bio-Rad Cat. # 170–8841) were added, and the reaction was incubated 5 min at 25℃, 30 min at 42℃ and 5 min at 85℃. cDNA was diluted 1:100 in water, and 5 µl were mixed with 5 µl of iTaq Universal SYBR Green Supermix (Bio-Rad Cat. # 172–5121) containing 400 nM each of forward and reverse primer (hAXIN2-RT-PCR-1-FOR: 5'-GTCCAG-CAAAACTCTGAGGG-3', hAXIN2-RT-PCR-1-REV: 5'-CTGGTGCAAAGACATAGCCA-3'; hCTNNB1-RT-PCR-1-FOR: 5'-AAAGCGGCTGTTAGTCACTGG-3', hCTNNB1-RT-PCR-1-REV: 5'-CGAGTCA TTGCATACTGTCCAT-3'; hHPRT1-RT-PCR-1-FOR: 5'-TGCTGAGGATTTGGAAAGGG-3', hHPRT1-RT-PCR-1-REV: 5'-ACAGAGGGCTACAATGTGATG-3'). Triplicate reactions for each cDNA and primer pair were prepared in a MicroAmp Optical 384-well Reaction Plate (Thermo Fisher Scientific Cat. # 4309849), sealed with MicroAmp Optical Adhesive Film (Thermo Fisher Scientific Cat. # 4311971) and run using standard parameters in an ABI 7900 Fast Real-Time PCR system (Applied Biosystems) controlled by the Sequence Detection Systems software version 2.4.1 provided by the manufacturer. The formula $2^{-\Delta Ct}$ was used to calculate the average relative abundance of *AXIN2* (*Figures 2A*, *6B and F* and *Figure 6—figure supplement 1A and E*) or *CTNNB1* (*Figure 6—figure supplement 1C*) mRNA normalized to *HPRT1* mRNA, and fold-changes in mRNA abundance were calculated as the quotient between the experimental and reference samples, with appropriate error propagation of the respective standard deviations (SD).

## Immunoblot analysis of HAP1-7TGP and mutant cell lines

Quantitative immunoblot analysis of soluble CTNNB1 from membrane-free supernatant (MFS) (*Figures 2C*, *3D,F*, *6C,D,G and H*, *Figure 3—figure supplement 1C* and *Figure 6—figure supplement 1B and F*)

Approximately 24 hr before treatment cells were seeded in 6 cm dishes at a density of $4 \times 10^6$ per dish and grown in 6 ml of CGM 2. Cells were treated for 10 hr with 50% WNT3A CM or 10 µM of the GSK3 inhibitor CHIR-99021 in CGM 2 where indicated. Cells were harvested, lysed by hypotonic shock, and extracts were prepared as follows, with all handling done on ice or at 4℃. Cells were washed twice with ~5 ml cold PBS and twice with ~5 ml cold 10 mM HEPES pH 7.4. Residual buffer was removed, and 100 µl of ice cold SEAT buffer (10 mM triethanolamine/acetic acid pH 7.6, 250 mM sucrose, 1X SIGMA*FAST* Protease Inhibitor Cocktail Tablets EDTA-free (Sigma-Aldrich Cat. # S8830), 25 µM MG132 (Sigma-Aldrich Cat. # C2211), 1X PhosSTOP (Roche Cat. # 04906837001), 1 mM NaF, 1 mM $Na_3VO_4$, 1 mM dithiothreitol (DTT), 62.5 U/ml Benzonase Nuclease (EMD Millipore Cat. # 70664), 1 mM $MgCl_2$) were added to the cells. Cells were scraped using a cell lifter (Corning Cat. # 3008), transferred to 2-ml centrifuge tubes and disrupted mechanically by triturating 10 times. Crude extracts were centrifuged for 20 min at 20,000 x g to pellet membranes and nuclei, and the membrane-free supernatant (MFS) was carefully removed, avoiding contamination from the pellet. The MFS was flash-frozen in liquid nitrogen and stored at −80℃ until further processing.

Extracts were thawed quickly at RT and transferred to ice. The protein concentration in the MFS was quantified with the Pierce BCA Protein Assay Kit (Thermo Fisher Scientific Cat. # 23225), using BSA as a standard, and samples were normalized by dilution with SEAT buffer. The MFS was diluted with 4X LDS sample buffer (Thermo Fisher Scientific Cat. # NP0007) supplemented with 50 mM *tris* (2-carboxyethyl)phosphine (TCEP), heated for 5 min at 95℃, and 30 µg of total protein were electrophoresed alongside Precision Plus Protein All Blue Prestained Protein Standards (Bio-Rad Cat. # 1610373) in NuPAGE 4–12% Bis-Tris gels (Thermo Fisher Scientific, various Cat. numbers) using 1X NuPAGE MOPS SDS running buffer (Thermo Fisher Scientific Cat. # NP0001), or on 8% Tris-polyacrylamide home-made gels using Tris-glycine SDS running buffer (25 mM Tris, 192 mM glycine, 0.1% SDS).

Proteins were transferred to nitrocellulose membranes in a Criterion Blotter apparatus (Bio-Rad Cat. # 1704071) using 1X NuPAGE transfer buffer (Thermo Fisher Scientific Cat. # NP0006) + 10% methanol for Bis-Tris gels, or Tris-glycine wet transfer buffer (12 mM Tris, 96 mM Glycine, 10% methanol) for Tris-polyacrylamide gels. Membranes were cut, blocked with Odyssey Blocking Buffer, incubated with mouse anti-$\beta$-catenin, mouse anti-GAPDH or mouse anti-ACTIN primary antibodies, washed with TBST, incubated with IRDye 800CW donkey anti-mouse IgG secondary antibody, washed with TBST followed by TBS, and imaged using the Li-Cor Odyssey imaging system. Acquisition parameters in the manufacturer's Li-Cor Odyssey Image Studio software were set so as to avoid saturated pixels in the bands of interest, and bands were quantified using manual background subtraction. The integrated intensity for CTNNB1 was normalized to that for GAPDH or ACTIN in the same blot, and the average ± SD from duplicate blots was used to represent the data, except for the TFAP4 experiment shown in *Figure 2C*, for which only one blot was analyzed.

## Other immunoblot analyses of MFS (*Figures 2C*, *6D and H* and *Figure 5— figure supplement 1A*)

The same membrane probed for GAPDH in the experiment shown in *Figure 2C* was subsequently incubated with rabbit anti-AP4 (TFAP4). The same membranes or duplicates of the ones probed for GAPDH in the experiments shown in *Figure 6D and H* were subsequently incubated with rabbit anti-SERBP1 or goat anti-casein kinase 1α primary antibodies, respectively. The same membrane probed for CTNNB1 in the experiment shown in *Figure 6D* was subsequently incubated with rabbit anti-APC. Following washes, the blots were incubated with IRDye 680RD donkey anti-rabbit IgG, peroxidase AffiniPure donkey anti-goat IgG, or peroxidase AffiniPure goat anti-rabbit IgG secondary antibodies, as appropriate. The TFAP4 and SERBP1 blots were imaged using the Li-Cor Odyssey imaging system, and the CSNK1A1 and APC blots were developed using the SuperSignal West Femto Maximum Sensitivity Substrate (Thermo Fisher Scientific Cat. # 34095).

To probe for HUWE1 (~480 kDa, *Figure 6H*) the same extracts used to quantify CTNNB1 in *Figure 6H* were electrophoresed in a 4% Tris-polyacrylamide gel strengthened with 0.5% agarose (based on the protocol of *Kinoshita et al., 2009*) using Tris-glycine SDS running buffer. Proteins were transferred to nitrocellulose membranes using Tris-glycine wet transfer buffer, blocked with TBST + 5% skim milk, and incubated with rabbit anti-Lasu1/Ureb1 (HUWE1) primary antibody, followed by peroxidase AffiniPure goat anti-rabbit IgG secondary antibody. The membrane was developed using the SuperSignal West Femto Maximum Sensitivity Substrate.

## Immunoblot analysis of total AXIN1 and AXIN2 (*Figure 3—figure supplement 1A*)

Approximately 24 hr before treatment, cells were seeded in 6 cm dishes at a density of $4 \times 10^6$ per dish, and grown in 6 ml of CGM 2. Cells were treated with 10 µM CHIR-99021 in CGM 2 for 14 hr to induce expression of *AXIN2*. Cells were harvested in 1 ml Trypsin-EDTA (0.05%), resuspended in 3 ml CGM 2, centrifuged at 400 x g for 5 min, washed in 1.9 ml PBS, transferred to 2-ml centrifuge tubes and centrifuged at 800 x g for 5 min. The supernatant was aspirated and the cell pellets were flash-frozen in liquid nitrogen and stored at −80°C. Pellets were thawed quickly at RT and transferred to ice. All subsequent steps were done on ice or at 4°C. The cell pellets were resuspended in 150 µl of ice-cold RIPA lysis buffer (50 mM Tris-HCl pH 8.0, 150 mM NaCl, 2% NP-40, 0.25% deoxycholate, 0.1% SDS, 1X SIGMA*FAST* protease inhibitors, 1 mM MgCl$_2$, 62.5 U/ml Benzonase Nuclease, 1 mM DTT, 10% glycerol), sonicated in a Bioruptor 300 (Diagenode) $4 \times 30$ s in the high setting, centrifuged 10 min at 20,000 x g and the supernatant was recovered.

The protein concentration in the supernatant was quantified using the Pierce BCA Protein Assay Kit. Samples were normalized by dilution with RIPA lysis buffer, further diluted with 4X LDS sample buffer supplemented with 50 mM TCEP, heated for 5 min at 95°C, and 100 µg of total protein were electrophoresed alongside Precision Plus Protein All Blue Prestained Protein Standards in NuPAGE 4–12% Bis-Tris gels using 1X NuPAGE MOPS SDS running buffer.

Proteins were transferred to nitrocellulose membranes using 1X NuPAGE transfer buffer + 10% methanol, and membranes were cut and blocked with TBST + 5% skim milk. Blots were incubated with goat anti-AXIN1, rabbit anti-AXIN2 or mouse anti-GAPDH primary antibodies, washed with

TBST, incubated with Peroxidase AffiniPure anti-goat, anti-rabbit or anti-mouse secondary antibodies, washed with TBST followed by TBS, and developed with SuperSignal West Femto.

## Immunofluorescence of HAP1-7TGP and mutant cell lines

### CTNNB1 immunostaining (*Figure 3—figure supplement 2B–2F*)

Approximately 24 hr before treatment, cells were seeded at a density of $4 \times 10^4$ per well on Poly-L-Lysine-coated #1 12 mm round coverslips placed in 24-well plates, and were grown in 0.5 ml of CGM 2. Cells were treated for 4 hr (or the periods indicated in the time-course experiment depicted in *Figure 3—figure supplement 2A*) with CGM 2 or with 50% WNT3A CM in CGM2 where indicated. Following treatment, the medium was removed, and cells were washed once with PBS and fixed with 4% paraformaldehyde in PBS for 10 min. Fixed cells were washed with PBS $3 \times 5$ min, blocked with blocking solution (PBS, 0.5% Triton X-100, 3% BSA) for 30 min and stained with mouse anti-$\beta$-catenin in blocking solution for 30 min at RT. Cells were washed with PBS $3 \times 5$ min and stained with donkey anti-mouse IgG (H+L) Alexa Fluor 647 conjugate in blocking solution for 30 min at RT. Cells were washed with PBS $3 \times 5$ min and mounted on slides using ProLong Diamond Antifade Mountant with DAPI (Thermo Fisher Scientific Cat. # P36962). Fluorescence micrographs were acquired on a Leica SP8 laser scanning confocal microscope equipped with a 63X oil immersion objective (NA 1.4) and an HyD hybrid detector, using Leica Application Suite X (LASX) software. During a given experiment, all Z-stacks were acquired with identical settings (laser power, gain, offset, frame format). Confocal sections were viewed and exported as TIF files using Volocity 6.3 (PerkinElmer), adjusted equally for contrast and brightness when necessary for clarity using Photoshop CS5 (Adobe), and arranged in Illustrator CS5 (Adobe).

### Quantification of nuclear CTNNB1 (*Figure 3E and F* and *Figure 3—figure supplement 2A*)

Leica Image Files (LIF) were converted into MATLAB matrices (MAT) in Volocity 6.3 (PerkinElmer). Matrices were manually curated to remove out-of-focus optical planes and analyzed using MATLAB R2014b (MathWorks) to determine the mean fluorescence intensity of nuclear CTNNB1 staining. Briefly, the fluorescence values from the DAPI channel were smoothed using a low-pass Gaussian filter to remove fine noise, and a threshold was applied to generate a nuclear mask. The nuclear mask was eroded uniformly to prevent contamination with cytoplasmic signal and optical sections with poor nuclear masks were discarded. The fluorescence values from the irrelevant pool of CTNNB1 at tight junctions were removed from the CTNNB1 channel in each plane using a high-pass Gaussian filter. The sum of the corrected fluorescence intensity values from the CTNNB1 channel co-localizing with the nuclear mask were divided by the nuclear mask area to obtain the mean nuclear fluorescence per unit area for each optical plane. The final reported value for each field of view, referred to as 'nuclear CTNNB1', is the median of the mean nuclear fluorescence per unit area obtained from the three most central optical sections. The graphs presented in *Figure 3E* and *Figure 3—figure supplement 2A* display the average (± SD) nuclear CTNNB1 from two or three fields of view.

### *Xenopus laevis* body axis duplication assay

mRNA was synthesized using the mMESSAGE mMACHINE SP6 Kit (Thermo Fisher Scientific Cat. # AM1340) from a pCSDest-based construct containing the CDS encoding the protein of interest. One microgram of plasmid DNA was linearized and mRNA was synthesized according to the protocol provided by the manufacturer. The mRNA was treated with TURBO DNase and purified using the RNeasy Mini Kit (QIAGEN Cat. # 74104), following the 'RNA cleanup' protocol. The purified product was analyzed on a 1% agarose gel. *X. laevis* eggs were fertilized, de-jellied with L-cysteine, and equilibrated in Marc's Modified Ringers (MMR: 0.1 M NaCl, 2 mM KCl, 1 mM $MgSO_4$, 2 mM $CaCl_2$, 5 mM HEPES pH 7.8, 0.1 mM EDTA) containing 2% Ficoll. One ventral blastomere of four-cell stage embryos was injected with 3 nl of a 1.67 ng/nl mRNA solution (5 ng total) using an MPPI-2 Pressure Injector (Applied Scientific Instrumentation). Embryos were incubated at RT in 1/3 MMR until stage 34 and scored for body axis duplication (*Figure 2F*) on a Zeiss Stemi 2000-C stereo microscope. Embryos were fixed overnight in MEMFA buffer (100 mM MOPS, 2 mM EGTA, 1 mM $MgSO_4$, 3.7% formaldehyde) transferred to 100% glycerol and images (*Figure 2E*) were taken with an Olympus DP72 camera at 2.5X magnification.

### *Drosophila melanogaster* methods

#### Flies and transgenes

The driver *C765-Gal4* (BDSC) was used for all crosses. All experiments were performed at 25°C. *pUASTattB-Axin-V5* was generated as described previously (*Yang et al., 2016*). To generate the *pUASTattB-AxinΔC-V5* transgene, residues 705 through 745 were deleted by PCR-based mutagenesis of *pUASTattB-Axin-V5* using the oligonucleotide 5'-GGTCACTTTAGGAAGCTTGGTAAACC-3'. The resulting *AxinΔC-V5* fragment was digested with KpnI and HindIII, and cloned into the *pUAS-TattB* vector digested with the same enzymes. Transgenic flies were generated using site-specific integration at the *attP33* site using phiC31-based integration (*Bischof et al., 2007*).

#### Imaging of adult fly wings (*Figure 3G*)

Images of adult fly wings were acquired with a Nikon E800 Epifluorescence microscope using Olympus DP software.

#### Immunostaining of wing imaginal discs (*Figure 3—figure supplement 3B*)

Third instar larval wing imaginal discs were dissected in PBS, fixed with 4% paraformaldehyde in PBS for 20 min, washed with PBS + 0.1% Triton X-100, and incubated in blocking solution (PBS, 0.5% Triton X-100, 10% BSA) for 1 hr. Imaginal discs were incubated with anti-Senseless or anti-Wingless primary antibodies in PBS + 0.5% Triton X-100 overnight at 4°C, and with appropriate secondary antibodies for 2 hr at RT. Fluorescence images were acquired using a Nikon A1RSi confocal microscope and processed using Photoshop CS5 (Adobe).

#### Immunostaining of embryos (*Figure 3—figure supplement 3C*)

Embryos were fixed in 3.7% formaldehyde (*Vorwald-Denholtz and De Robertis, 2011*) and rehydrated in PBT (PBS, 1% BSA, 0.1% Tween-20). Following incubation for 1 hr in blocking solution (PBS, 10% BSA, 0.1% Tween-20), embryos were incubated with mouse anti-Engrailed/Invected or mouse anti-Wingless primary antibodies in PBT overnight at 4°C. After washing with PT (PBS, 0.1% Tween-20), embryos were incubated for 1 hr with donkey anti-mouse IgG (H+L) Alexa Fluor 555 secondary antibody for 1 hr at RT. Embryos were then washed with PT and mounted in ProLong Gold Antifade Mountant (Thermo Fisher Scientific Cat. # P10144). Fluorescence images were acquired on a Nikon NIS confocal microscope, processed using Photoshop CS5 (Adobe) and assembled using Illustrator CS5 (Adobe).

#### Immunoblotting of embryonic lysates (*Figure 3—figure supplement 3E*)

Embryos were lysed in lysis buffer (50 mM Tris-HCl pH 8.0, 100 mM NaCl, 1% NP-40, 10% glycerol, 1.5 mM EDTA pH 8.0, 1X Halt Protease and Phosphatase Inhibitor Cocktail (Thermo Fisher Scientific Cat. # 78440)). Lysates were fractionated on 8% SDS-PAGE, transferred to PVDF membranes and blocked in TBST + 5% skim milk. Membranes were incubated with mouse anti-V5 or mouse anti-α-Tubulin primary antibodies, washed with TBST, incubated with anti-mouse IgG HRP conjugated secondary antibody, washed, and developed using ECL reagent (250 mM luminol, 90 mM p-coumaric acid, 100 mM Tris-HCl pH 8.5, 30% hydrogen peroxide). Protein levels were quantified by densitometry of scanned film using Image J software (National Institutes of Health).

#### Cuticle preparations (*Figure 3—figure supplement 3G*)

Larval cuticles were prepared as described previously (*Wieschaus and Nüsslein-Volhard, 1986*).

#### Embryonic hatch rate determination (*Figure 3—figure supplement 3H*)

Fly eggs collected during a 1 hr period were incubated at 25°C for an additional 24 hr before the hatch rate was scored.

### RNAseq analysis

Gene expression levels in WT HAP1 cells were determined from previously published data (*Dubey et al., 2016*). Briefly, the raw RNAseq data for WT HAP1 cells (NCBI Gene Expression Omnibus (GEO) Series accession number GSE75515) was aligned, quantified and analyzed using Partek

Flow software, version 4.0 (Partek Inc.). Reads were aligned to human reference genome build hg19 using the STAR Align and Quantify pipeline in Partek Flow. RPKM (Reads Per Kilobase of transcript per Million mapped reads) normalization was used to obtain the values reported in *Table 1*. The normalized data was imported into Partek Genomics Suite 6.6 software (Partek Inc.) to visualize the quantification of normalized reads for selected genes, and *Table 1* was assembled in Excel (Microsoft).

## Preparation of figures and statistical analysis

Illustrations were prepared using PowerPoint (Microsoft) and Illustrator CS6 (Adobe). Circle plots depicting the hits from each screen, GT insertion histograms, dose response graphs, tables and supplementary files were prepared using Excel (Microsoft). Bar and circle graphs were prepared using Prism 6 (GraphPad Software) and statistical analysis was performed using the same software (details of statistical tests used are given in the figure legends). Heat maps were generated using Partek Genomics Suite 6.6 software (Partek Inc.) and finished in Illustrator CS6. FACS histograms and dot plots were generated using FlowJo (FlowJo, LLC) and finished in Illustrator CS6. Pictures of immunoblots and model organisms were only adjusted for contrast and brightness when necessary for clarity using Photoshop CS6 (Adobe), and were arranged in Illustrator CS6.

## Acknowledgements

We thank Marc Kirschner, Timothy Mitchison, Tom Rapoport, Henry Ho, Lingyin Li and Ganesh Pusapati for comments on the manuscript. We thank Marc Kirschner, Timothy Mitchison, Tom Rapoport, Roel Nusse and members of the Rohatgi and Carette labs for input on the project. We thank Yan Ma for help making constructs and cell lines. We thank Patty Lovelace and Jennifer Cheung for training and assistance with FACS, and the Stanford Institute for Stem Cell Biology and Regenerative Medicine FACS Core for use of instruments, purchased in part using an S10 Shared Instrumentation Grant (1S10RR029333801) from the National Institutes of Health (NIH). This work was funded in part by NIH grants DP2 GM105448 (RR), DP2 AI104557 (JEC), R01 GM081635 (EL), R01 GM103926 (EL) and RO1 CA105038 (YA), by the Norris Cotton Cancer Center (YA) and by start-up funds from the Stanford Cancer Institute (RR). JEC is a David and Lucile Packard Foundation fellow and RR is a Josephine Q. Berry Faculty Scholar in Cancer Research at Stanford. AML was supported by the Stanford Dean's Postdoctoral Fellowship, the Stanford Cancer Biology Program Training Grant and the Novartis sponsored Fellowship from the Helen Hay Whitney Foundation. The authors declare no conflicts of interest.

## Additional information

### Funding

| Funder | Grant reference number | Author |
| --- | --- | --- |
| Helen Hay Whitney Foundation | Novartis Fellowship | Andres M Lebensohn |
| National Institutes of Health | RO1 CA105038 | Yashi Ahmed |
| National Institutes of Health | R01 GM081635 | Ethan Lee |
| National Institutes of Health | R01 GM103926 | Ethan Lee |
| National Institutes of Health | DP2 AI104557 | Jan E Carette |
| David and Lucile Packard Foundation | Fellow Award | Jan E Carette |
| Stanford University School of Medicine | Josephine Q.Berry Fellowship | Rajat Rohatgi |
| National Institutes of Health | DP2 GM105448 | Rajat Rohatgi |

The funders had no role in study design, data collection and interpretation, or the decision to submit the work for publication.

## Author contributions

AML, Conceived the study, Designed experiments, Analyzed the data, Conducted or oversaw all experiments except those in model organisms, Wrote the manuscript; RD, Executed the CSNK1A1 suppressor screen, Gave feedback on the manuscript; LRN, Conducted *X. laevis* experiments; OT-B, EY, Conducted *D. melanogaster* experiments; CDM, Prepared sequencing libraries; EMD, BBP, Developed computational tools for the comparative analysis of screens; ZB-N, Executed the APC suppressor screen; KJT, Imaged and quantified nuclear CTNNB1 in haploid human cells; YA, Oversaw *D. melanogaster* experiments, Gave feedback on the manuscript; EL, Oversaw *X. laevis* experiments, Gave feedback on the manuscript; JEC, Conceived the study, Designed experiments, Analyzed the data, Gave feedback on the manuscript; RR, Conceived the study, Designed experiments, Analyzed the data, Wrote the manuscript

## Author ORCIDs

Andres M Lebensohn, http://orcid.org/0000-0002-4224-8819
Rajat Rohatgi, http://orcid.org/0000-0001-7609-8858

# Additional files

### Supplementary files

• Supplementary file 1. Ranked lists of hits from all screens. Genes containing at least one GT insertion in the population of sorted cells from each genetic screen described in this work are listed in separate spreadsheets (the screen name is indicated on the tab of each spreadsheet), and are ranked based on the significance of GT insertion enrichment (*p*-value) in the sorted vs. the unsorted (control) cells. For the unsorted cells, the number of GT insertions in genes (regardless of orientation) is indicated for the complete dataset and for each gene (column B). For the sorted cells, the number of inactivating GT insertions in genes (sense and antisense insertions in exons and sense insertions in introns), as well as the number of sense or antisense GT insertions in exons or in introns, is indicated for the complete dataset and for each gene (columns C-G). Three measures of GT insertion enrichment are shown: the *p*-value and the FDR-corrected *p*-value (both derived from columns B and C), and the Intronic GT Insertion Orientation Bias (IGTIOB) score (derived from columns F and G). See Materials and methods for details.

• Supplementary file 2. List of clonal cell lines used in this study. Clones in which a single gene was targeted using CRISPR/Cas9 or disrupted by a GT insertion, and double- or triple-mutant clones in which multiple genes were disrupted using CRISPR/Cas9 or through a combination of CRISPR/Cas9 and a GT insertion are described in two separate spreadsheets labeled accordingly. For cell lines engineered using CRISPR/Cas9, when more than one clone was generated using the same CRISPR guide, the 'Clone Name' column indicates the generic name used throughout the manuscript to describe the genotype, and the 'Clone #' column identifies the specific allele in each individual clone. The 'CRISPR guide' column indicates the name of the guide used, which is the same as that of the oligos encoding sgRNAs (see Materials and methods and *Supplementary file 4*). The 'Genomic Sequence' column shows 80 bases of genomic sequence (5' relative to the gene is to the left) surrounding the target site. For each group of clones made using the same CRISPR guide (separated by gray spacers), the 'Genomic Sequence' column is headlined by the reference WT genomic sequence (obtained from RefSeq), with the guide sequence colored blue. The site of the double strand cut made by Cas9 is between the two underlined bases. Sequencing results for individual clones are indicated below the reference sequence. Some WT clones are indicated as such and were used as controls. For mutant clones, mutated bases are colored red (dashes represent deleted bases, three dots are used to indicate that a deletion continues beyond the 80 bases of sequence shown, and large insertions are indicated in brackets), and the nature of the mutation, the resulting genotype (with the mutated amino acid number specified in parenthesis for selected clones) and any pertinent observations are also described. The figures in which each clone was used are also indicated. For the APC$^{KO-2}$ clone containing a GT insertion, the 80 bases of genomic sequence (5' relative to the gene is to the left) flanking the GT are shown in the 'Genomic Sequence' column. For double- or triple- mutant clones, the CRISPR guide or pair of guides used (if two different guides

were used simultaneously to target adjacent sites in the same gene), the genomic sequence, the mutation, the genotype and any observations pertaining to each of the two or three targeted genes are designated '1', '2' and '3' in the column headings, and are shown under green, orange and purple spacers, respectively.

• Supplementary file 3. Lists of the most significant hits included in comparative analyses across screens. Genes used to generate the heat map (*Figure 4B*) comparing the two WNT positive regulator screens (*Figure 1E and F*) and the low WNT + RSPO screen (*Figure 4A*) are shown in the first spreadsheet, and those used to generate the heat map (*Figure 5D*) comparing the WNT positive regulator, low stringency screen (*Figure 1E*) and the APC and CSNK1A1 suppressor screens (*Figure 5B and C*) are shown in the second spreadsheet. Genes with an FDR-corrected $p$-value$<10^{-4}$ in at least one of the three screens included in each comparison are shown, and the FDR-corrected $p$-value and IGTIOB score for each gene in each screen is indicated. Genes are shown in the same order as in the heat maps, clustered based on their IGTIOB scores (see Materials and methods for details).

• Supplementary file 4. List of oligonucleotides and primers used for generation and characterization of clonal cell lines engineered using CRISPR/Cas9. The names and sequences of pairs of oligonucleotides encoding sgRNAs (which were cloned into pX330) are shown in columns A and B, respectively. The names and sequences of pairs of primers used to amplify corresponding genomic regions flanking sgRNA target sites are shown in columns C and D, respectively. The names and sequences of single primers used to sequence the amplified target sites are shown in columns E and F, respectively.

### Major datasets

The following dataset was generated:

| Author(s) | Year | Dataset title | Dataset URL | Database, license, and accessibility information |
|---|---|---|---|---|
| Lebensohn AM, Dubey R, Neitzel LR, Tacchelly-Benites O, Yang E, Marceau CD, Davis EM, Patel BB, Bahrami-Nejad Z, Travaglini KJ, Ahmed Y, Lee E, Carette JE, Rohatgi R | 2016 | Comparative genetic screens in human cells reveal new regulatory mechanisms in WNT signaling | https://trace.ncbi.nlm.nih.gov/Traces/sra/?study=SRP094861 | Publicly available at the NCBI Sequence Read Archive website (accession no: SRP094861) |

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
