## [Decision Letter]

Thank you for submitting your article "Comparative genetic screens in human cells reveal new regulatory mechanisms in WNT signaling" for consideration by *eLife*. Your article has been favorably evaluated by Tony Hunter as the Senior Editor and three reviewers, one of whom, Utpal Banerjee (Reviewer #1), is a member of our Board of Reviewing Editors.

The reviewers are unanimous in their support of the work; they have discussed the reviews with one another. The changes suggested are minor and can be addressed by a few changes in the text. The Reviewing Editor has drafted this decision to help you prepare a revised submission. We hope you will be able to submit the revised version within two months.

Summary:

Lebensohn et al. have carried out a genome wide screen for novel effectors of Wnt signaling. Instead of siRNA-, overexpression- or genetic screens used previously, they used mutagenesis in haploid cells and an elegant analysis approach. Conducting the screen under various activation modes permitted to screen for select classes of activators/inhibitors. Via limited validation they provide evidence for new factors TFAP4, GPI-anchor biosynthetic pathway (for R-spondin signaling), SERBP1. In addition, their work identified C-terminal truncation mutants for Axin2, which act as dominant negatives and whose epistasis analysis suggest additional layers of β-catenin regulation, which is independent from its degradation. They identified HUWE1, a known negative Wnt regulator, is shown to also have a positive mode of action.

Essential revisions:

1) Please further comment on limitations of the approach due to inherent redundancy of gene function, and perhaps how future screens could address these issues.

2) The Axin story is confusing. The authors condense the description of why the allele is dominant negative, and need to be clearer on what was learnt from this allele. In particular, the reasons for inactive β-catenin accumulation are not properly developed. The authors should either describe this better, or perhaps consider whether this part is essential to their overall story.

3) The selection is based on a GFP reporter carrying 7 TCF binding elements, which is unlikely to faithfully and fully mimic endogenous regulatory sequences downstream of Wnt. Please add a comment on how this might affect interpretation of results, particularly those pertaining to hits in genes that encode proteins involved in chromatin modification. The authors should discuss follow-ups that allow measurement of endogenous gene activity.

---

## [Author Response]

*Essential revisions:*

*1) Please further comment on limitations of the approach due to inherent redundancy of gene function, and perhaps how future screens could address these issues.*

We have now explicitly addressed the fact that our approach would miss redundant or essential genes in the Results section. This limitation is inherent to all “forward” genetic screens that use random or untargeted mutagenesis. It is possible that CRISPR/Cas9 screens, in which the genomic location of the mutations is chosen by the guide RNA, could be better suited to address issues of redundancy, for example by simultaneously mutating multiple members of a gene family.

*2) The Axin story is confusing. The authors condense the description of why the allele is dominant negative, and need to be clearer on what was learnt from this allele. In particular, the reasons for inactive β-catenin accumulation are not properly developed. The authors should either describe this better, or perhaps consider whether this part is essential to their overall story.*

We have made significant changes to the text that describes the AXIN section of the paper to clarify the logic of the experiments and the consequent conclusions (subsection “The C-terminal DAX domain of AXIN2 controls CTNNB1 transcriptional activity”). The discussion in this section has been made more concise to focus on the new information gained by the isolation of this allele, as suggested above. We have included a direct comparison of our results with current models of DAX-domain function from the literature (see aforementioned subsection) to highlight the fact that this allele uncovers an unanticipated role for AXIN in regulating β-catenin transcriptional activity downstream of the destruction complex (and hence independent of β-catenin protein levels). The precise biochemical mechanism behind why β-catenin remains inactive when the DAX domains is deleted will require significant additional investigation (though we have ruled out a defect in nuclear translocation in Figure 3E,F and Figure 3—Figure supplement 2), which we believe is outside the scope of this already expansive manuscript.

We have retained this section of the paper (with substantial modifications) because the AXIN story makes two important contributions to the overall message of the manuscript. First, it moves the WNT signaling field forward by demonstrating an unexpected biochemical function for the DAX domain conserved from flies to humans. Second, it highlights one of the unique advantages of using untargeted mutagenesis in haploid cells over the more commonly used CRISPR/Cas9 screening system, in which mutations are targeted to specific regions of genes by the guide RNA. Indeed, our work suggests that it might be possible to isolate rare (and often informative) dominant alleles by constructing guide RNA libraries that systematically target the whole exome.

*3) The selection is based on a GFP reporter carrying 7 TCF binding elements, which is unlikely to faithfully and fully mimic endogenous regulatory sequences downstream of Wnt. Please add a comment on how this might affect interpretation of results, particularly those pertaining to hits in genes that encode proteins involved in chromatin modification. The authors should discuss follow-ups that allow measurement of endogenous gene activity.*

We have now explicitly addressed this point in the second paragraph of the subsection “Forward genetic screens in haploid human cells identify negative, attenuating, and positive regulators of WNT signaling”, commenting on how reporter genes and endogenous target genes may significantly differ in their sensitivity to chromatin modifications. We also note in this section that we have addressed this concern by confirming new regulatory mechanisms (whenever possible) by measuring the expression of endogenous target genes (Figures 2A, 6B, F, Figure 6—Figure supplement 1A and E), measuring the entirely non-transcriptional readout of β-catenin stabilization (Figures 3D,F, 6C,D,G,H, Figure 3—Figure supplement 1C, Figure 6—Figure supplements 6B and 6F), or assessing WNT-driven phenotypes in animals (Figures 2E,F, 3G,H, Figure 3—Figure supplement 3), rather than relying solely in reporter gene activity data. Finally, we have not pursued the detailed analysis of any screen hits that encode chromatin-modifying enzymes.